# ATR limits Rad18-mediated PCNA monoubiquitination to preserve replication fork and telomerase-independent telomere stability

Siyuan Chen [1,2], Chen Pan[2], Jun Huang [1,2,3]✉ & Ting Liu [1,4]✉

## Abstract

Upon replication fork stalling, the RPA-coated single-stranded DNA (ssDNA) formed behind the fork activates the ataxia telangiectasia-mutated and Rad3-related (ATR) kinase, concomitantly initiating Rad18-dependent monoubiquitination of PCNA. However, whether crosstalk exists between these two events and the underlying physiological implications of this interplay remain elusive. In this study, we demonstrate that during replication stress, ATR phosphorylates human Rad18 at Ser403, an adjacent residue to a previously unidentified PIP motif (PCNA-interacting peptide) within Rad18. This phosphorylation event disrupts the interaction between Rad18 and PCNA, thereby restricting the extent of Rad18-mediated PCNA monoubiquitination. Consequently, excessive accumulation of the tumor suppressor protein SLX4, now characterized as a novel reader of ubiquitinated PCNA, at stalled forks is prevented, contributing to the prevention of stalled fork collapse. We further establish that ATR preserves telomere stability in alternative lengthening of telomere (ALT) cells by restricting Rad18-mediated PCNA monoubiquitination and excessive SLX4 accumulation at telomeres. These findings shed light on the complex interplay between ATR activation, Rad18-dependent PCNA monoubiquitination, and SLX4-associated stalled fork processing, emphasizing the critical role of ATR in preserving replication fork stability and facilitating telomerase-independent telomere maintenance.

**Keywords** ATR; RAD18; PCNA Monoubiquitination; Fork Collapse; Telomere Stability
**Subject Categories** DNA Replication, Recombination & Repair; Post-translational Modifications & Proteolysis

## Introduction

Genome duplication involves the intricate coordination of multiple replication forks originating from dispersed origins, meticulously orchestrated to ensure efficient and accurate replication (Burgers and Kunkel, 2017; O'Donnell et al, 2013; Sclafani and Holzen, 2007). Despite this coordination, replication forks continually encounter various challenges, including DNA lesions arising from both endogenous and exogenous sources, as well as intrinsic obstacles such as complex DNA sequences, tightly bound protein–DNA complexes, and conflicts between replication and transcription (Ciccia and Elledge, 2010; Debatisse et al, 2012; García-Muse and Aguilera, 2016; Zeman and Cimprich, 2014). These challenges can lead to disruptions and impediments in replication fork progression, compromising replication fidelity and efficiency, and ultimately contributing to genome instability and human diseases (Burrell et al, 2013; Macheret and Halazonetis, 2015; Zeman and Cimprich, 2014).

One common consequence of DNA replication slowdown or blockage is the formation of extensive stretches of single-stranded DNA (ssDNA), rapidly bound by the replication protein A (RPA) complex, due to the uncoupling of replicative polymerase and helicase movements (Chen et al, 2013; Deng et al, 2015; Pacek and Walter, 2004). This ssDNA-RPA complex signals the activation of the ATR kinase, a pivotal effector in the response to replication stress (Cortez et al, 2001; Zou and Elledge, 2003). Once activated, ATR phosphorylates numerous substrates, resulting in cell-cycle arrest, replisome stabilization, and preservation of stalled replication forks to prevent them from collapsing into DNA double-strand breaks (DSBs) (Branzei and Foiani, 2009; Friedel et al, 2009; Saldivar et al, 2017). ATR employs various mechanisms to prevent fork collapse, including the global inhibition of origin firing to maintain an adequate RPA pool for ssDNA protection during stress (Toledo et al, 2013). Failure to suppress late-origin firing depletes the RPA pool and triggers a replication catastrophe (Dungrawala et al, 2015; Toledo et al, 2013). Additionally, ATR orchestrates the stabilization and remodeling of stalled replication forks through various downstream regulators and effectors (Chanoux et al, 2009; Couch et al, 2013; Koundrioukoff et al, 2013; Lopes et al, 2001;

[1]Zhejiang Provincial Key Laboratory of Geriatrics and Geriatrics Institute of Zhejiang Province, Affiliated Zhejiang Hospital, Zhejiang University School of Medicine, 310058 Hangzhou, China. [2]The MOE Key Laboratory of Biosystems Homeostasis & Protection and Zhejiang Provincial Key Laboratory of Cancer Molecular Cell Biology, Life Sciences Institute, Zhejiang University, 310058 Hangzhou, China. [3]Center for Life Sciences, Shaoxing Institute, Zhejiang University, 321000 Shaoxing, China. [4]Department of Cell Biology, and Department of General Surgery of Sir Run Run Shaw Hospital, Zhejiang University School of Medicine, 310058 Hangzhou, China. ✉E-mail: jhuang@zju.edu.cn; liuting518@zju.edu.cn

Tercero and Diffley, 2001). For instance, ATR phosphorylates the DNA translocase SMARCAL1 to restrain its activity in fork regression (Couch et al, 2013). Without this phosphorylation, SMARCAL1 generates DNA structures susceptible to cleavage by the SLX4 structure-specific endonuclease complexes, resulting in the formation of DSBs (Couch et al, 2013; Fekairi et al, 2009; Forment et al, 2011; Hanada et al, 2007). However, the precise mechanisms governing SLX4 recruitment to stalled forks remain elusive.

Proliferating cell nuclear antigen (PCNA) plays a pivotal role as a processivity factor in eukaryotic DNA replication, serving as a central scaffold that orchestrates the dynamic and precise engagement of numerous factors within the replication machinery (Georgescu et al, 2014; Georgescu et al, 2015; Wang et al, 2016; Zheng and Shen, 2011). Furthermore, PCNA functions as a versatile docking platform, facilitating the recruitment of essential components involved in the replication surveillance mechanisms (Bienko et al, 2005; Guo et al, 2006; Kannouche et al, 2004; Plosky et al, 2006; Watanabe et al, 2004). The interaction between PCNA and its partner proteins primarily relies on the well-defined PCNA-interacting peptide (PIP) box (Prestel et al, 2019). The regulated and tightly coordinated interplay between PCNA and these effector proteins constitutes a central regulatory hub in pathways responding to replication stress (Choe and Moldovan, 2017; Moldovan et al, 2007). These pathways are governed by intricate, multi-layered regulatory mechanisms encompassing posttranslational modifications that affect both PCNA and its associated proteins (Gali et al, 2012; Geng et al, 2010; Hoege et al, 2002; Papouli et al, 2005). Intriguingly, ATR activation and Rad18-dependent PCNA monoubiquitination are both triggered by RPA-coated ssDNA during replication stress in a similar manner (Cortez et al, 2001; Davies et al, 2008; Zou and Elledge, 2003). However, the precise molecular crosstalk between ATR activation and Rad18-dependent PCNA monoubiquitination, as well as the broader physiological implications of this interplay, remain unresolved.

In this study, we unveiled a novel mechanism wherein ATR phosphorylates human Rad18 at Ser403 during replication stress, disrupting its interaction with PCNA. This regulatory step curtails Rad18-mediated PCNA monoubiquitination, thereby preventing the excessive accumulation of SLX4 at stalled forks and, consequently, stalling fork collapse. Notably, ATR's role extends to maintaining telomere stability in ALT cells by limiting Rad18-mediated PCNA ubiquitination and excessive SLX4 accumulation at telomeres. These findings shed light on ATR's critical contribution to preserving replication fork stability and facilitating telomerase-independent telomere maintenance through Rad18 phosphorylation.

# Results

## ATR restricts Rad18-mediated PCNA monoubiquitination in response to replication stress

To investigate the potential involvement of ATR in PCNA monoubiquitination induced by replication stress, we treated cells with hydroxyurea (HU), a ribonucleotide reductase inhibitor known to induce replication stress, alone or in combination with the pharmacological ATR inhibitor VE-821 (verified by Chk1

Ser345 phosphorylation). Consistent with previous studies (Davies et al, 2008), short-term treatment with HU resulted in a modest increase in the levels of monoubiquitinated PCNA (Fig. 1A). Intriguingly, concurrent treatment with VE-821 significantly augmented PCNA ubiquitination (Fig. 1A), suggesting a potential negative regulatory role of ATR. This observation was further supported by the knockdown of ATR using small interfering RNA (siRNA), which also significantly enhanced the HU-induced monoubiquitination of PCNA (Fig. 1B).

To validate and expand these findings, we assessed whether ATR inhibition could potentiate PCNA monoubiquitination induced by other replication stress-causing treatments. As shown in Fig. 1C,D, ATR inhibition similarly led to a substantial increase in mono-ubiquitinated PCNA levels when fork stalling resulted from aphidicolin, a DNA polymerase inhibitor, or exposure to ultraviolet (UV) radiation. Importantly, the downregulation of human Rad18, the primary E3 ligase responsible for PCNA monoubiquitination, largely abolished PCNA monoubiquitination induced by combined treatment with HU and ATRi (Fig. 1E). Taken together, these results suggest that ATR may constrain Rad18-mediated PCNA monoubiquitination under conditions of replication stress.

Upon activation, ATR activates its downstream kinases, including Chk1 and Wee1, to inhibit cell-cycle progression, suppress late-origin firing, and stabilize stalled forks (Lee et al, 2001; O'Connell et al, 1997; Saldivar et al, 2017; Zhu et al, 2023). To elucidate whether ATR directly inhibits replication stress-induced monoubiquitination of PCNA, or exerts its impact indirectly through its downstream kinases, we simultaneously treated cells with HU and inhibitors of Chk1 or Wee1. As shown in Fig. 1F, neither UCN-01 nor MK-8776, inhibitors of Chk1 (evaluated through Chk1 Ser296 autophosphorylation), nor MK-1775, an inhibitor of Wee1 (assessed by CDK1 Tyr15 phosphorylation), augmented HU-induced PCNA monoubiquitination, suggesting that ATR may directly impede PCNA monoubiquitination in response to replication stress, independently of its downstream effectors. Furthermore, the application of PHA-767491, a CDC7 inhibitor (validated by MCM2 Ser40 phosphorylation) widely used to suppress origin firing, was incapable of abrogating the effects caused by ATR inhibition (Fig. 1G), ruling out the possibility that unscheduled origin firing contributes to the increased PCNA monoubiquitination in cells treated with HU and ATRi.

## Excessive Rad18-mediated PCNA monoubiquitination resulting from ATR inhibition contributes to stalled fork collapse

ATR inhibition triggers the collapse of stalled replication forks, leading to the formation of DSBs and subsequent accumulation of single-stranded DNA (ssDNA), which facilitates DSB repair through homologous recombination (Couch et al, 2013). To elucidate the potential relationship between ATR inhibition-induced PCNA monoubiquitination and fork collapse, we conducted a temporal analysis of PCNA monoubiquitination and ATM phosphorylation, a well-established marker of DSBs, following drug treatment. As shown in Fig. 2A, PCNA monoubiquitination reached its peak at 30 min upon co-treatment with HU and ATRi, while ATM phosphorylation continued to increase even at 120 min. This temporal relationship suggests that the excessive monoubiquitination of PCNA may precede the breakage of stalled forks,

potentially contributing to their collapse. To validate this hypothesis, we employed Rad18 depletion and observed a significant, albeit incomplete, reduction in ATM phosphorylation in cells treated with HU and ATRi (Fig. 2B). Importantly, the inhibitory effect of Rad18 knockdown on fork breakage in HU and ATRi-treated cells was largely restored by re-expression of siRNA-resistant wild-type Rad18 (Fig. 2B). Furthermore, reintroduction of siRNA-resistant wild-type PCNA, but not the ubiquitination-resistant K164R mutant, effectively counteracted the suppressive effect of PCNA depletion on ATM phosphorylation (considering the essential role of PCNA in cell proliferation, only partially

knockdown of PCNA was performed) (Fig. 2C). Collectively, these results suggest that Rad18-mediated monoubiquitination of PCNA at lysine 164 stimulates the breakage of stalled forks, with ATR playing a crucial role in suppressing this detrimental consequence of replication stress.

To further explore the role of excessive Rad18-mediated PCNA monoubiquitination in this process, we conducted a DNA fiber assay. In this assay, HeLa cells were labeled with the modified thymidine analog iododeoxyuridine (IdU), and replication was blocked by treating them with high concentrations of HU in the presence or absence of ATRi. Subsequently, the drugs were

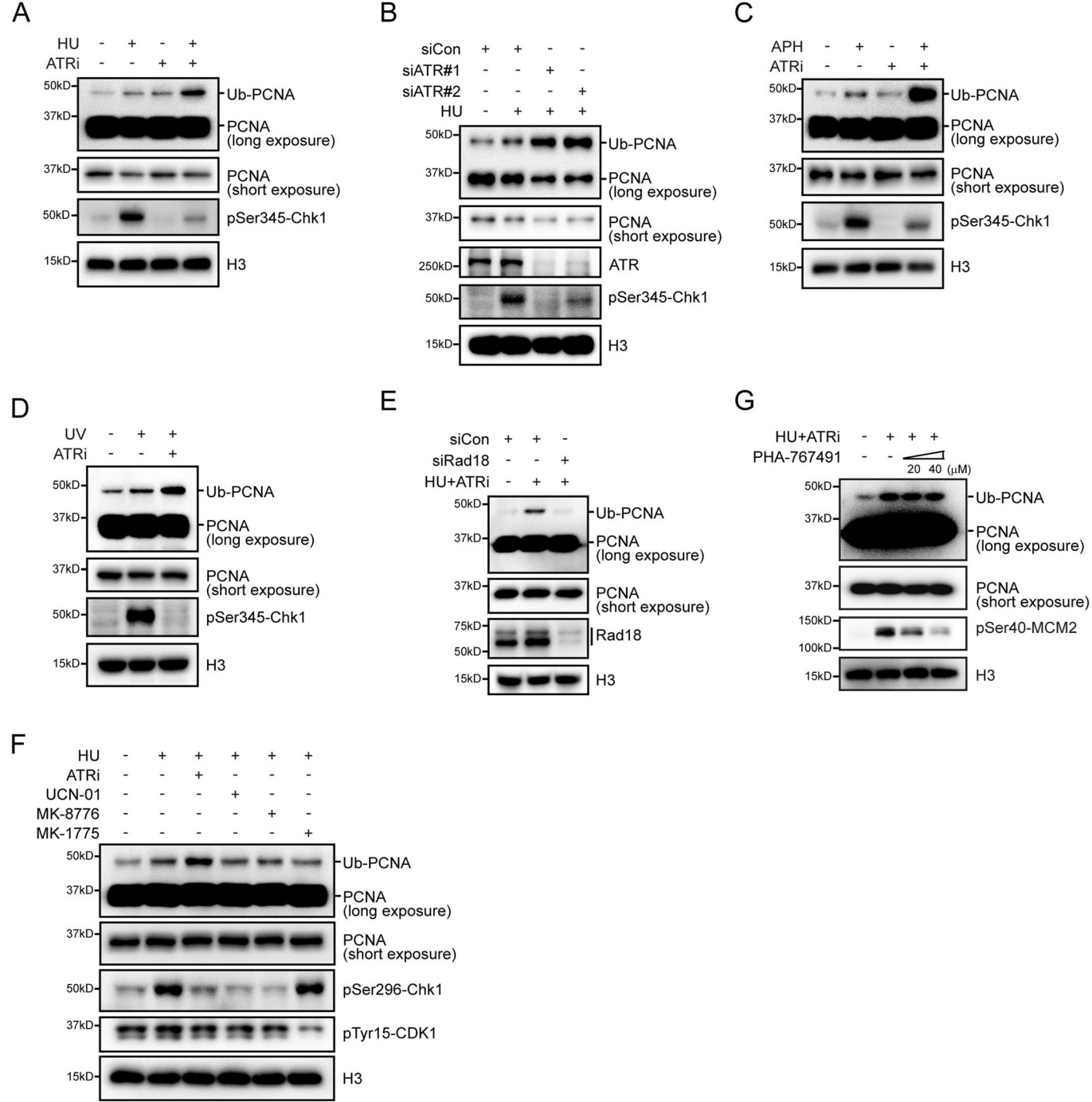

**Figure 1.   ATR inhibition enhances Rad18-mediated PCNA ubiquitination during replication stress.**

(A) Stimulation of PCNA ubiquitination under replication stress upon ATR inhibition. HeLa cells were either mock-treated with DMSO or exposed to 2 mM HU, 2 μM VE-821, or a combination of 2 mM HU and 2 μM VE-821 for 1 h. Chromatin fractions were isolated and immunoblotted using the specified antibodies. (B) ATR depletion promotes PCNA ubiquitination under replication stress. HeLa cells were transfected with ATR-specific siRNAs. After 48 h of transfection, cells were treated with 2 mM HU for 1 h. Chromatin fractions were isolated and immunoblotted using the specified antibodies. (C) HeLa cells were either mock-treated with DMSO or exposed to 20 μM APH, 2 μM VE-821, or a combination of 20 μM APH and 2 μM VE-821 for 1 h. Chromatin fractions were isolated and immunoblotted using the specified antibodies. (D) HeLa cells were either mock-treated with DMSO, exposed to 30 J/m² UV, or treated with a combination of 30 J/m² UV and 2 μM VE-821 for 1 h. Chromatin fractions were isolated and immunoblotted using the specified antibodies. (E) Rad18 mediates PCNA ubiquitination under replication stress upon ATR inhibition. HeLa cells were transfected with Rad18-specific siRNAs. After 48 h of transfection, cells were treated with a combination of 2 mM HU and 2 μM VE-821 for 1 h. Chromatin fractions were isolated and immunoblotted using the specified antibodies. (F, G) ATR downstream effects are dispensable for heightened PCNA ubiquitination. HeLa cells were treated with combinations of 2 mM HU and ATR inhibitor VE-821 (2 μM), Chk1 inhibitor UCN-01 (300 nM)/MK-8776 (5 μM), Wee1 inhibitor MK-1775 (10 μM), CDC7 inhibitor PHA-767491 (20 or 40 μM), or an equivalent volume of DMSO as indicated for 1 h. Chromatin fractions were isolated and immunoblotted using the specified antibodies. Source data are available online for this figure.

removed, and the cells were labeled with a second thymidine analog chlorodeoxyuridine (CldU) (Fig. 2D). As shown in Figure EV1A,B, treatment with ATR inhibitor resulted in a substantial decrease in the ratio of CldU to IdU track lengths, indicating that ATR inactivation hampered the restart of stalled replication forks. In line with this, ATR inhibition significantly increased the number of forks that failed to restart and completely collapsed during the HU treatment, as evidenced by the absence of CldU incorporation after release (Fig. 2D–F). Notably, short-term ATR inhibition did not obviously induce fork degradation under replication stress (Figure EV1C,D). Remarkably, the depletion of Rad18 reduced the frequency of collapsed forks, approaching the level observed with the depletion of SLX4, a well-known factor implicated in facilitating fork collapse (Fig. 2E,F). Consistently, the depletion of Rad18 also largely reversed the defects in fork restart caused by ATR inhibition (Fig. EV1A,B). More importantly, the simultaneous depletion of Rad18 and SLX4 did not cause a further increase in the ratio of CldU to IdU track lengths (Figure EV1E,F), indicating a functional interdependency between Rad18 and SLX4 within the same pathway.

We next tested whether the excessive Rad18-mediated PCNA monoubiquitination resulting from ATR inhibition would also contribute to the generation of ssDNA at stalled forks. To this end, we performed a native BrdU labeling assay. HeLa cells were subjected to BrdU labeling for 15 min and then exposed to a combination of 2 mM HU and 2 μM VE-821 for 3 h. Under non-denaturing conditions, the BrdU antibody specifically detects ssDNA. Consistent with previous findings (Couch et al, 2013), treatment with a dimethylsulfoxide (DMSO) vehicle showed minimal BrdU staining, indicating limited ssDNA formation (Fig. 2G–I). In contrast, co-treatment with HU and ATRi resulted in robust BrdU staining, indicating the presence of ssDNA at the nascent DNA strands (Fig. 2G–I). Strikingly, depletion of Rad18, although not completely, significantly reduced the formation of BrdU foci (Fig. 2G–I), suggesting a significant dependency on Rad18-mediated PCNA monoubiquitination for the generation of ssDNA. Notably, the decrease in BrdU foci formation observed in Rad18-depleted cells was fully restored by reintroducing siRNA-resistant wild-type Rad18 (Fig. 2G–I).

It was demonstrated that ZRANB3, a DNA translocase, interacts with polyubiquitinated PCNA to facilitate replication fork reversal (Vujanovic et al, 2017). To examine the significance of PCNA polyubiquitination-dependent ZRANB3-mediated fork reversal in fork collapse following HU and ATRi treatment, we depleted ZRANB3 and assessed its impact on ATM phosphorylation and

BrdU foci formation. As shown in Fig. EV1G–I, ZRANB3 knockdown did not significantly affect ATM phosphorylation or BrdU foci formation in cells treated with HU and ATRi. These results suggest that PCNA polyubiquitination-dependent ZRANB3-mediated fork reversal may not be a prerequisite for the observed fork collapse under these conditions.

The aforementioned results raised the possibility that Rad18 depletion may confer resistance to ATRi treatment in cells experiencing replication stress. As shown in Fig. 2J, Rad18 depletion indeed increased the resistance of HeLa cells to ATRi treatment under conditions of HU-induced replication stress. To further explore this, we analyzed a pharmacogenomics dataset comprising 1001 human cancer cell lines (Iorio et al, 2016) and examined the correlation between half maximal inhibitory concentration (IC50) values of VE-821 and Rad18 gene expression levels. This analysis was conducted on 890 cell lines for which both gene expression data and drug responses to VE-821 were available. The statistical analysis revealed a significant negative association between Rad18 expression and VE-821 IC50 values (Fig. 2K). Collectively, these results underscore the crucial role of regulated PCNA monoubiquitination mediated by ATR in preserving genomic stability during replication stress.

## ATR phosphorylates human Rad18 at Ser403 in response to replication stress

Upon a thorough examination of the amino acid sequence of human Rad18, we identified a conserved ataxia telangiectasia-mutated (ATM)/ATR phosphorylation consensus motif (SQ/TQ, amino acids 403–404) within its C-terminal region (Fig. 3A). Considering the existing evidence indicating ATR's role in regulating Rad18-mediated PCNA monoubiquitination during replication stress, we hypothesized that Rad18 might directly serve as a substrate for ATR. To test this hypothesis, we conducted immunoprecipitation of both endogenous and SFB-tagged Rad18 following induction of replication stress using HU. Western blot analysis using an anti-phospho-Ser403-Rad18 antibody unveiled specific phosphorylation of Rad18 in HU-treated cells (Fig. 3B,C). This phosphorylation coincided with Chk1 phosphorylation, a recognized marker of replication stress (Fig. 3B). Importantly, the observed Rad18 phosphorylation signal was reliant on ATR, evident as treatment with the ATR inhibitor not only diminished the Rad18 phosphorylation but also abrogated Chk1 phosphorylation (Fig. 3B). Furthermore, substituting Ser403, situated within the SQ/TQ consensus motif of Rad18, with either alanine or glutamic

                                                                               

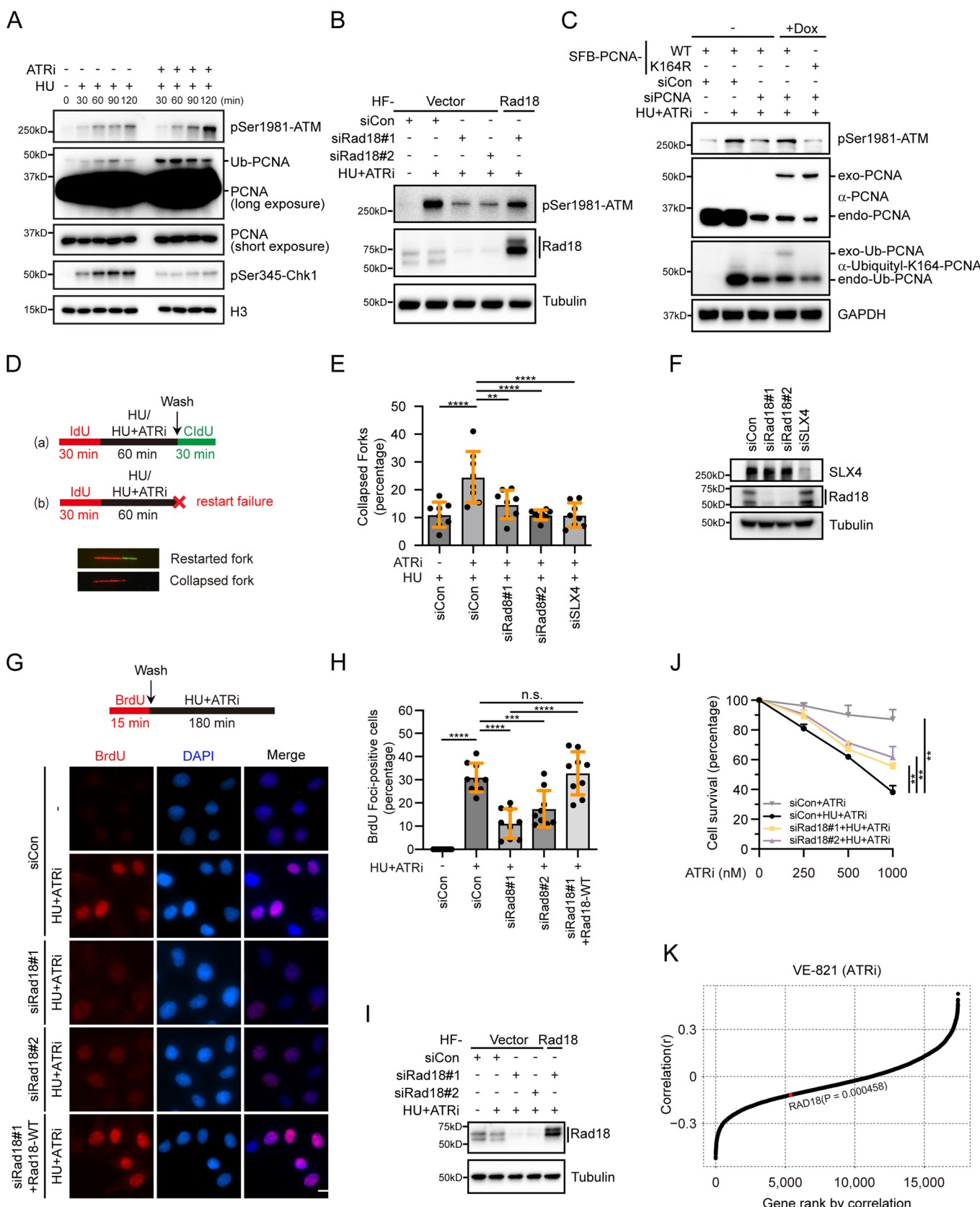

**Figure 2.   ATR-dependent inhibition of PCNA ubiquitination prevents fork collapse.**

(A) Kinetics of ATM phosphorylation and PCNA ubiquitination. HeLa cells treated with 2 mM HU alone or in combination with 2 µM VE-821 were collected at indicated time points. Chromatin fractions were isolated for PCNA ubiquitination detection by immunoblotting using specific antibodies. (B) Rad18-mediated PCNA ubiquitination promotes fork collapse. A stable HeLa cell line expressing HA-Flag (HF)-tagged siRNA#1-resistant Rad18 or an empty vector was generated. The resulting cell lines were transfected with the indicated siRNAs. After 48 h of transfection, cells were treated with 2 mM HU and 2 µM VE-821 for 3 h. Cell lysates were then prepared and western blot analysis was carried out as indicated. (C) A HeLa cell line expressing siRNA-resistant SFB-tagged wild-type PCNA or the K164R mutant under the control of a tetracycline-inducible promoter was generated. The resulting cells were transfected with control siRNA or siRNA against PCNA. 24 h post-transfection, cells were treated with doxycycline (1 µg/mL) to induce the expression of PCNA. 48 h later, cells were either mock-treated or treated with 2 mM HU and 2 µM VE-821 for 3 h. Cell lysates were then prepared and western blot analysis was carried out as indicated. Endo-PCNA refers to endogenous PCNA, while exo-PCNA indicates exogenous PCNA. Monoubiquitination of PCNA was detected using a Ubiquityl-Lysine 164-PCNA specific antibody. (D) Schematic of the DNA fiber experiment. HeLa cells were transfected with indicated siRNAs. 48 h after transfection, cells were incubated with 50 µM IdU for 30 min, treated with 2 mM HU and 2 µM VE-821 for 1 h, and incubated with 100 µM CldU for 30 min. Bottom: representative images of a restarted fork and a collapsed fork (Red-only tracks). (E) Rad18 depletion prevents fork collapse. Quantification of collapsed forks in cells transfected with the indicated siRNAs. Data represent means ± SD of three independent experiments. Each black dot in the graph represents the percentage of collapsed forks in each measurement, and more than 200 fibers were measured for each sample. **P < 0.01; ****P < 0.0001, one-way ANOVA test. (F) Western blot analysis of knockdown efficiency of Rad18 and SLX4. (G) Rad18 is required for nascent-strand ssDNA generation at stalled replication forks when ATR is inhibited. Top: Schematic of the nascent ssDNA detection via native BrdU immunofluorescence assay. A HeLa cell line stably expressing HF-tagged siRNA#1-resistant Rad18 or an empty vector was transfected with the indicated siRNAs. After 48 h of transfection, cells were labeled with 10 µM BrdU for 15 min and then either mock-treated or treated with 2 mM HU and 2 µM VE-821. After 3 h, cells were fixed and stained with an antibody against BrdU for nascent-strand ssDNA detection without DNA denaturation. Bottom: Representative BrdU foci in cells transfected with the indicated siRNAs. Scale bar, 10 µm. (H) Quantification of BrdU foci. Cells with more than five BrdU foci were considered positive. Data represent means ± SD of three independent experiments. Each black dot in the graph represents the percentage of BrdU-positive cells in each measurement, with more than 300 cells were counted in each experiment. ***P < 0.001; ****P < 0.0001, n.s. indicates not significant, determined by a one-way ANOVA test. (I) Western blot analysis of Rad18 expression. (J) Rad18 depletion confers cellular resistance to ATR inhibitor. HeLa cells transfected with indicated siRNAs were treated with various doses of VE-821 (0, 250, 500, 1000 nM) with or without 50 µM HU for 10 days before staining. The results shown are means of three independent experiments and are presented as means ± SEM. **P < 0.01, determined by t-tests. (K) Correlation between the IC50 values of VE-821 and Rad18 mRNA levels. Two-sided P values were determined using the relation between the estimated coefficient and the Student's t-distribution. Source data are available online for this figure.

acid completely abolished the phosphorylation signal (Fig. 3C). These results strongly suggest that ATR phosphorylates Rad18 at Ser403 in response to replication stress. It is worth noting that a previous proteomic study also identified Rad18 as a potential target of ATM/ATR (Matsuoka et al, 2007).

## Ser403 and its adjacent PIP motif within human Rad18 mediate its interaction with PCNA

Upon replication stress, Rad18 associates with RPA and is recruited to stalled forks, where it facilitates the monoubiquitination of PCNA (Davies et al, 2008). To investigate the significance of ATR-mediated phosphorylation of Rad18 at Ser403, we examined its effect on the interaction between Rad18 and RPA or PCNA. For this purpose, we purified His-sumo-tagged wild-type human Rad18, along with the phosphorylation-deficient S403A mutant, and the phosphorylation-mimic mutant S403E, from *E. coli* and performed pulldown experiments (the sumo tag improved Rad18's solubility, enabling expression and purification without RAD6) (Hibbert et al, 2011). As shown in Fig. 3D, both the S403A and S403E mutants pulled down endogenous RPA1 from HEK293T cell extracts, similar to wild-type Rad18, indicating that the Ser403 residue is dispensable for Rad18's interaction with RPA. However, while the recombinant wild-type Rad18 efficiently pulled down endogenous PCNA from HEK293T cell extracts as well as recombinant PCNA purified from *E. coli*, the phosphorylation-mimic mutant S403E failed to do so (Fig. 3D,E), indicating that phosphorylation at Rad18 Ser403 may impede its association with PCNA. Surprisingly, even when serine 403 was mutated to alanine, which was expected to retain binding ability similar to wild-type Rad18 due to the absence of phosphorylation, the interaction with PCNA remained less efficient (Fig. 3D,E). Similar results were obtained with co-immunoprecipitation experiments (Fig. 3F). These results suggest that the Ser403 residue of Rad18 is critical

for its interaction with PCNA, and changes in this residue, including phosphorylation modification, may induce conformational alterations in an unidentified motif necessary for Rad18's association with PCNA.

Considering the established mode of interaction between PCNA-associated proteins and PCNA via their respective PIP (PCNA-interacting peptide) boxes, we explored the presence of a potential PIP motif near serine 403, which could be influenced by phosphorylation. Through a visual inspection of the human Rad18 amino acid sequence, we identified a potentially degenerate PIP box (QXXL, amino acids 404-407) situated immediately adjacent to serine 403 (Fig. 3A). To assess whether Rad18 indeed interacts with PCNA via this putative PIP box, we conducted pull-down and co-immunoprecipation experiments using wild-type Rad18 and a PIP motif deletion mutant (deletion spanning from glutamine 404 to serine 409, Rad18-ΔPIP). As shown in Fig. 3F,G, deletion of the PIP motif resulted in reduced binding to PCNA. Moreover, mutations introduced within the consensus PIP box (substituting glutamine 404 and leucine 407 with alanine, Rad18-PIP-DM) also attenuated the Rad18-PCNA interaction, corroborating the significance of the PIP motif in the Rad18-PCNA interaction (Fig. 3G). Importantly, in contrast to the effect observed with wild-type Rad18, the overexpression of the S403 A mutant, S403E mutant, or PIP motif deletion mutant failed to enhance PCNA monoubiquitination in cells treated with HU and ATRi (Fig. 3H). These findings collectively suggest that both the Ser403 residue and its adjacent PIP motif within human Rad18 are critical for its interaction with PCNA, and that ATR-mediated phosphorylation at Ser403 restricts the Rad18–PCNA interaction, as well as subsequent PCNA monoubiquitination in response to replication stress.

Intriguingly, while the PIP motif is highly conserved across various species, the conservation of the serine 403 residue is limited to higher primates (Fig. 3A), implying that ATR-mediated Rad18 phosphorylation at Ser403 might be specific to higher primate

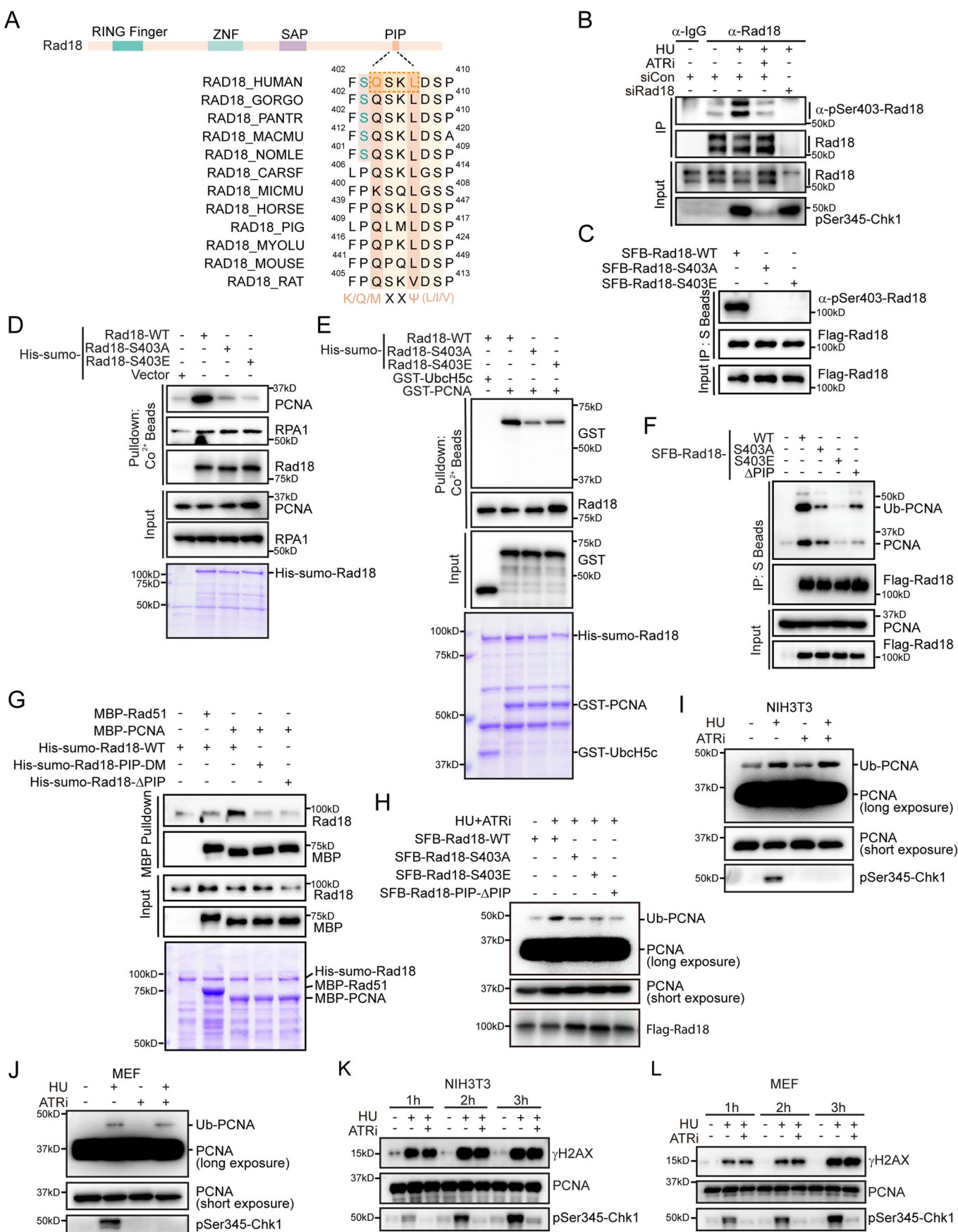

**Figure 3. ATR phosphorylates human Rad18 at Serine 403 to inhibit its association with PCNA.**

(A) Sequence alignment of serine 403 along with the PIP motif within Rad18 among different species. (B) ATR phosphorylates human Rad18 at serine 403 under replication stress. HeLa cells were treated with a combination of 2 mM HU and 2 μM VE-821 or an equivalent volume of DMSO for 1 h. Cell lysates were subjected to immunoprecipitation with Protein A beads conjugated with indicated antibodies. The phosphorylation signal was detected by western blot with a phospho-serine 403-Rad18-specific antibody. (C) HeLa cells were transfected with plasmids encoding SFB-tagged wild-type Rad18, the S403A or S403E mutants. Cell lysates were subjected to immunoprecipitation with S beads. The phosphorylation signal was detected by western blot with a phospho-serine 403-Rad18-specific antibody. (D, E) Phosphorylation of serine 403 on human Rad18 inhibits its association with PCNA. Purified His-sumo-tagged wild-type Rad18 or the indicated mutants were incubated with cell lysate from HEK293T (D) or recombinant GST-PCNA protein purified from *E. coli* (E). Purified proteins were visualized by Coomassie staining. (F) HEK293T cells were transfected with plasmids encoding SFB-tagged wild-type Rad18 or the indicated mutants. 24 h later, cells were lysed in NETN buffer and subjected to immunoprecipitation with S beads. Western blot analysis was carried out as indicated. (G) Serine 403, together with the PIP motif, is responsible for PCNA binding. Purified MBP-PCNA were incubated with His-sumo-tagged wild-type Rad18 or the indicated mutant proteins purified from *E. coli*. MBP-Rad51 served as a negative control. Immunoblotting was performed using indicated antibodies. (H) HeLa cells transfected with the indicated plasmids were treated with 2 mM HU and 2 μM VE-821 for 1 h. Chromatin fractions were isolated and immunoblotted with indicated antibodies. (I, J) Murine Cell Lines, NIH-3T3 or MEF cells, were treated with 2 mM HU alone or a combination of 2 mM HU and 2 μM VE-821 or an equivalent volume of DMSO for 1 h. Chromatin fractions were isolated and immunoblotted with indicated antibodies. (K, L) ATR inhibition does not accelerate fork collapse in murine cells. Murine cell lines, NIH-3T3 or MEF cells, treated with 2 mM HU alone or a combination of 2 mM HU and 2 μM VE-821 or an equivalent volume of DMSO, were collected at indicated time points and immunoblotted with indicated antibodies. Source data are available online for this figure.

organisms. To substantiate this notion, we examined the regulation of PCNA monoubiquitination by ATR in mouse cell lines, specifically MEF and NIH-3T3. As shown in Fig. 3I–L, inhibiting ATR did not further enhance replication stress-induced PCNA monoubiquitination and fork collapse in both cell lines. This observation strengthens the notion of higher primate-specific ATR-mediated regulation of PCNA monoubiquitination through Ser403 phosphorylation in Rad18.

## ATR restrains excessive SLX4 accumulation at stalled forks by phosphorylating Rad18

The SLX4–endonuclease complex generates DSBs at stalled replication forks when ATR activity is suppressed (Couch et al, 2013; Fekairi et al, 2009; Forment et al, 2011; Hanada et al, 2007). We thus hypothesized that inhibiting ATR could lead to abnormal SLX4 accumulation at stalled forks, a result of increased Rad18-mediated PCNA monoubiquitination. To test this hypothesis, we first employed a proximity ligation assay (PLA) combined with EdU labeling to quantitatively evaluate the localization of SLX4 at newly synthesized DNA. HeLa cells were labeled with EdU for 15 min, followed by treatment with HU alone or in combination with ATRi for 1 h. Subsequently, click chemistry was employed to biotinylate EdU, and PLA was conducted to visualize the colocalization of SLX4 with biotinylated EdU. As shown in Fig. 4A,B, ATR inhibition significantly increased the number of HU-induced SLX4/biotin PLA foci, indicating enhanced SLX4 accumulation at stalled forks. The specificity of the SLX4 antibody was verified by a dramatic reduction in PLA signal upon SLX4 depletion using SLX4-specific siRNAs.

We next investigated whether the excessive accumulation of SLX4 at stalled forks resulting from ATR inhibition relied on Rad18-mediated PCNA monoubiquitination. As shown in Fig. 4C,D, depletion of Rad18 substantially reduced the formation of SLX4/biotin PLA foci in cells treated with HU and ATRi. Remarkably, the inhibitory effect of Rad18 depletion on excessive SLX4 accumulation was largely rescued by reintroducing siRNA-resistant wild-type Rad18 (Fig. 4E,F). However, this rescue was not achieved with the phosphorylation-deficient mutants S403A and S403E, nor with the PIP motif deletion mutant (Fig. 4E,F). Consistently, the inhibitory effect of Rad18 depletion on ATM phosphorylation was largely restored by siRNA-resistant wild-type

Rad18, but not by the S403A and S403E mutants, or the PIP motif deletion mutant (Fig. 4G). Moreover, the simultaneous depletion of Rad18 and SLX4 did not induce a further decrease in ATM phosphorylation compared to individual depletions (Fig. 4H). Collectively, these findings indicate that ATR suppresses SLX4-dependent fork collapse at stalled replication forks, at least in part, by attenuating excessive PCNA monoubiquitination through Rad18 phosphorylation at Ser403.

SLX4 operates as a scaffold and engages with at least three different structure-selective endonucleases: XPF-ERCC1, SLX1, and MUS81-EME1 (Fekairi et al, 2009; Munoz et al, 2009; Svendsen et al, 2009). Given that XPF has been demonstrated to be responsible for the rapid breakage of replication forks induced by replication stress (Betous et al, 2018), we investigated its potential involvement in SLX4-dependent stalled fork collapse upon ATR inhibition. As shown in Fig. EV2A,B, HU stimulation led to the accumulation of XPF at stalled forks; however, ATR inhibition did not further increase this enrichment. Additionally, in contrast to SLX4 depletion, the absence of XPF did not suppress ATM phosphorylation or the frequency of fork collapse following HU and ATRi treatment (Fig. EV2C,D). Furthermore, despite MUS81 being responsible for the formation of DSBs after prolonged exposure to replication inhibitors (Hanada et al, 2007), we found that depleting MUS81 had no effect on ATM phosphorylation following HU and ATRi treatment (Fig. EV2E). Strikingly, the depletion of SLX1 significantly suppressed ATM phosphorylation under the same conditions, suggesting that the SLX1 endonuclease may play a key role in SLX4-dependent stalled fork collapse following ATR inhibition (Fig. EV2F).

## SLX4 recognizes monoubiquitinated PCNA via its UBZ4-1 domain to promote fork collapse

The N-terminal region of SLX4 harbors two consecutive ubiquitin-binding zinc finger 4 (UBZ4) domains (Fig. 5A). Based on our finding that excessive SLX4 accumulation at stalled replication forks relies on Rad18 following ATR inhibition, we postulated that SLX4 may specifically recognizes ubiquitinated PCNA through its tandem UBZ4 domains. To test this hypothesis, we introduced mutations in the conserved cysteine residues within the tandem UBZ4 domains of SLX4, either individually or in combination, and assessed their interaction with PCNA. As shown in Fig. 5B, treatment with HU and

ATRi enhanced the interaction between wild-type SLX4 and ubiquitinated PCNA. Intriguingly, this interaction was significantly reduced when the conserved cysteine residues within UBZ4-1 domain, but not within the UBZ4-2 domain, were mutated (Fig. 5B). Considering that a substantial amount of non-ubiquitinated PCNA co-precipitates with SLX4 regardless of replication stress (Fig. 5B), we speculated that SLX4 may contain a potential PIP box mediating its basal affinity for PCNA. Upon scrutinizing the amino acid sequence of the SLX4 protein, we identified three potential PIP boxes in the N-terminal region of SLX4 (Fig. 5A). Strikingly, deletion of the first (amino acid 58-59) or second (amino acid 204–216) putative PIP box, but not the third (amino acid 267–277), significantly disrupted the interaction between SLX4 and PCNA (Fig. 5C). These results suggest that both the UBZ4-1 domain and the PIP boxes may specifically mediate the interaction between SLX4 and the ubiquitinated PCNA under conditions of replication stress.

To ascertain the existence of a direct interaction between SLX4-UBZ4 and ubiquitinated PCNA, we produced and purified a

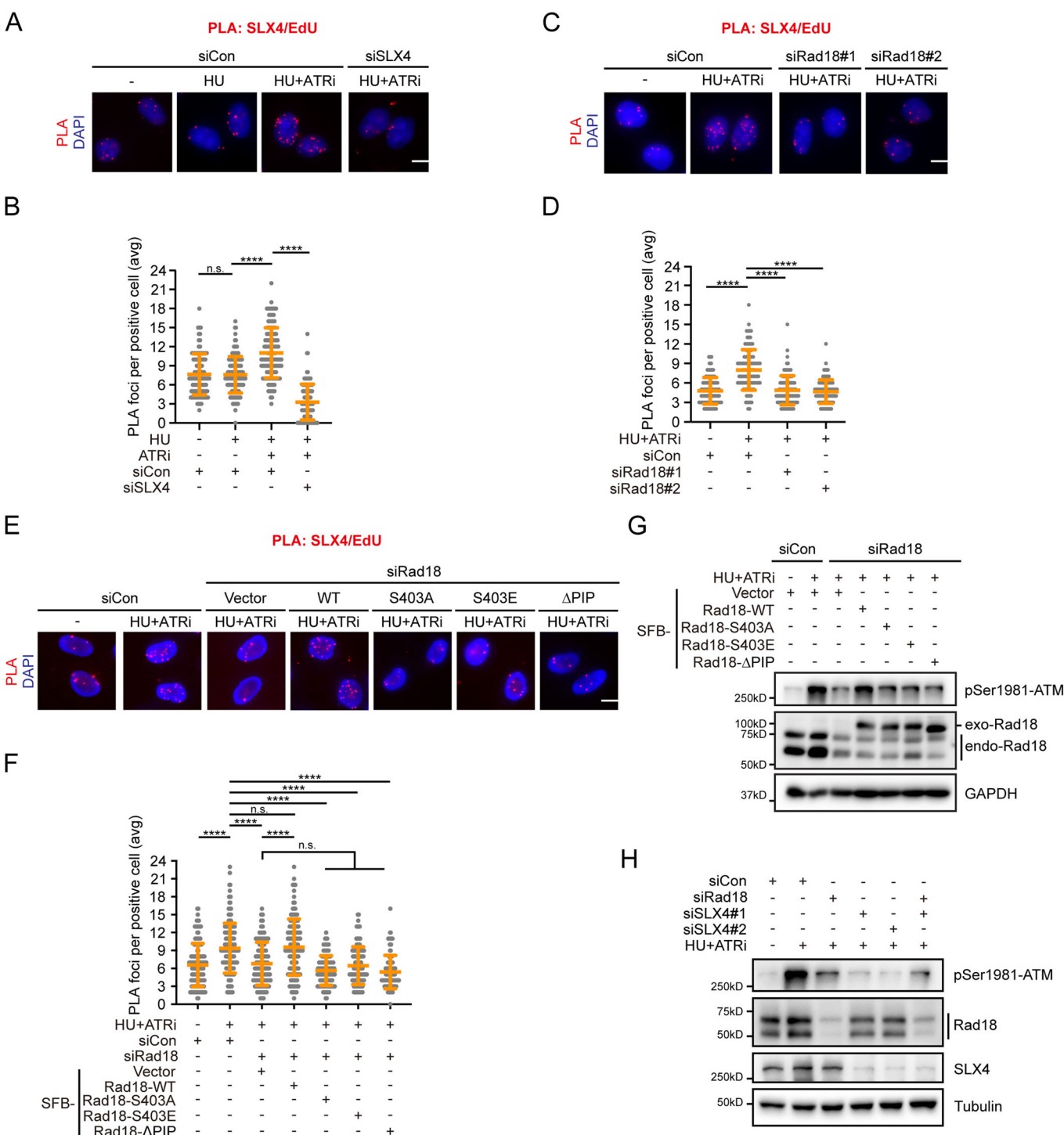

**Figure 4. ATR restrains excessive SLX4 accumulation at stalled replication forks.**

(A, B) HeLa cells were pulse-labeled with 10 μM EdU for 15 min, mock-treated, treated with 2 mM HU, or a combination of 2 mM HU and 2 μM VE-821 for 1 h, and then subjected to PLA with anti-SLX4 and anti-biotin antibodies. Representative images of PLA foci (red) (A). DNA was stained with DAPI. Scale bar, 10 μm. Quantification of PLA foci number per focus-positive cell (B). Data represent means ± SD of three independent experiments. More than 100 cells were counted for each sample. ****$P < 0.0001$, n.s., not significant, one-way ANOVA test. (C, D) Impaired SLX4 recruitment to stalled forks upon Rad18 depletion. HeLa cells were transfected with the indicated siRNAs, pulse-labeled with 10 μM EdU for 15 min, mock-treated or treated with a combination of 2 mM HU and 2 μM VE-821 for 1 h, and then subjected to PLA with anti-SLX4 and anti-biotin antibodies. Representative images of PLA foci (red) (C). DNA was stained with DAPI. Scale bar, 10 μm. Quantification of PLA foci number per focus-positive cell (D). Data represent means ± SD of three independent experiments. More than 100 cells were counted for each sample. ****$P < 0.0001$, n.s., not significant, one-way ANOVA test. (E, F) HeLa cells were transfected with the indicated siRNAs/plasmids, pulse-labeled with 10 μM EdU for 15 min, mock-treated or treated with a combination of 2 mM HU and 2 μM VE-821 for 1 h, and then subjected to PLA with anti-SLX4 and anti-biotin antibodies. Representative images of PLA foci (red) (E). DNA was stained with DAPI. Scale bar, 10 μm. Quantification of PLA foci number per focus-positive cell (F). Data represent means ± SD of three independent experiments. More than 100 cells were counted for each sample. ****$P < 0.0001$, n.s., not significant, one-way ANOVA test. (G) HeLa cells transfected with the indicated siRNAs/plasmids were mock-treated or treated with the combination of 2 mM HU and 2 μM VE-821 for 3 h. Immunoblotting was performed using antibodies as indicated. (H) HeLa cells were transfected with siRNAs as indicated. 48 h after transfection, cells were mock-treated or treated with a combination of 2 mM HU and 2 μM VE-821 for 3 h. Immunoblotting was performed using antibodies as indicated. Source data are available online for this figure.

GST-tagged SLX4 fragment containing the tandem UBZ4 domains (amino acids 292–361, referred to as GST-SLX4-UBZ4-WT) in *E. coli* (Fig. 5A). Additionally, we generated ubiquitinated SFB-PCNA from HEK293T cells (Fig. 5D). As shown in Fig. 5E, in vitro pulldown experiments confirmed the direct interaction between UBZ4-WT and monoubiquitinated PCNA. Notably, consistent with the co-immunoprecipitation findings, mutation of the UBZ4-1 domain, but not the UBZ4-2 domain, disrupted the direct interaction between SLX4-UBZ4 and monoubiquitinated PCNA (Fig. 5E). Correspondingly, the SLX4-C296A/C299A mutant that lacks the ability to bind monoubiquitinated PCNA failed to efficiently enrich at nascent DNA following HU and ATRi treatment (Fig. 5F,G). More importantly, the suppressive effect of SLX4 depletion on DSB formation following ATR inhibition was substantially reversed by the re-expression of wild-type SLX4, but not by the SLX4-C296 A/C299A mutant (Fig. 5H). Taken together, these results suggest that ATR restrains excessive SLX4 accumulation at stalled forks, thus preventing their collapse.

## ATR restrains excessive SLX4 accumulation at telomeres in ALT cells

Apart from its localization at DNA damage sites and stalled replication forks, SLX4 also localizes to PML bodies and interacts specifically with telomeric DNA within ALT-associated PML bodies (APBs) (Sobinoff et al, 2017; Wan et al, 2013). The antagonistic relationship between SLX4 and BLM, crucial for maintaining ALT activity at a productive but tolerable level (Sobinoff et al, 2017; Zhang and Zou, 2020), along with the disruptive effect of ATR inhibition on ALT activity (Flynn et al, 2015), suggests a potential regulatory role for ATR in the recruitment of SLX4 at ALT telomeres to promote ALT. Since ALT DNA synthesis primarily occurs during the G2 phase of the cell cycle (Zhang et al, 2019), we investigated whether ATR could also regulate PCNA monoubiquitination in G2 phase ALT-positive cells. To address this, we synchronized ALT-positive U2OS cells in the G2 phase using thymidine and the CDK1 inhibitor RO-3306 (Fig. 6A). Subsequently, the synchronized cells were treated with HU and ATRi while maintaining RO-3306 in the culture medium to prevent entry into the M phase (Fig. 6A). As shown in Fig. 6B, similar to the S phase, HU treatment alone modestly increased PCNA monoubiquitination during the G2 phase. Intriguingly, co-treatment with the ATR inhibitor significantly enhanced the monoubiquitination of PCNA (Fig. 6B,C).

To explore whether ATR inhibition-induced PCNA monoubiquitination leads to excessive SLX4 accumulation at ALT telomeres, we employed PLA to visualize the colocalization of SLX4 with TRF1. As shown in Fig. 6D,E, co-treatment with HU and ATRi substantially increased the number of SLX4/TRF1 PLA foci, specifically in the G2 phase but not in the G1 or S phase in ALT-positive U2OS cells, indicating enhanced accumulation of SLX4 at telomeres. In contrast, co-treatment with HU and ATRi had no effect on the number of SLX4/TRF1 PLA foci in ALT-negative HeLa cells (Fig. EV3A,B). Importantly, depletion of Rad18 significantly attenuated the formation of SLX4/TRF1 PLA foci in U2OS cells, suggesting the dependence of excessive SLX4 accumulation at telomeres resulting from ATR inhibition on Rad18-mediated PCNA monoubiquitination (Fig. 6F,G).

## ATR preserves telomere stability in ALT cells by restricting Rad18-mediated PCNA ubiquitination

The observations described above led us to hypothesize that ATR might regulate ALT activity by restricting Rad18-mediated PCNA monoubiquitination, thereby preventing excessive accumulation of SLX4 at telomeres in ALT cells. To test this hypothesis, we quantified changes in ALT activity by assessing the abundance of C-circles, a specific and quantifiable biomarker of ALT (Cesare and Griffith, 2004; Henson et al, 2009; Nabetani and Ishikawa, 2009), in U2OS cells. In accordance with previous findings (Flynn et al, 2015), inhibition of ATR led to a significant decrease in C-circle levels (Fig. 6H–J). Notably, the inhibitory effect of ATR inhibition on C-circle formation was completely abrogated upon depletion of either SLX4 or Rad18 (Fig. 6H–J). More importantly, the simultaneous depletion of Rad18 and SLX4 did not cause a further increase in the C-circle level (Fig. 6H–J), underscoring the pivotal roles of Rad18 and SLX4 functioning in the same pathway in mediating the impact of ATR on C-circle formation.

# Discussion

Despite substantial research efforts, a comprehensive understanding of how the ATR signaling pathway regulates DNA replication and safeguards stalled replication forks from collapse remains elusive. In our study, we unveil a novel ATR-dependent mechanism that critically protects replication forks in human cells, preserving genome integrity

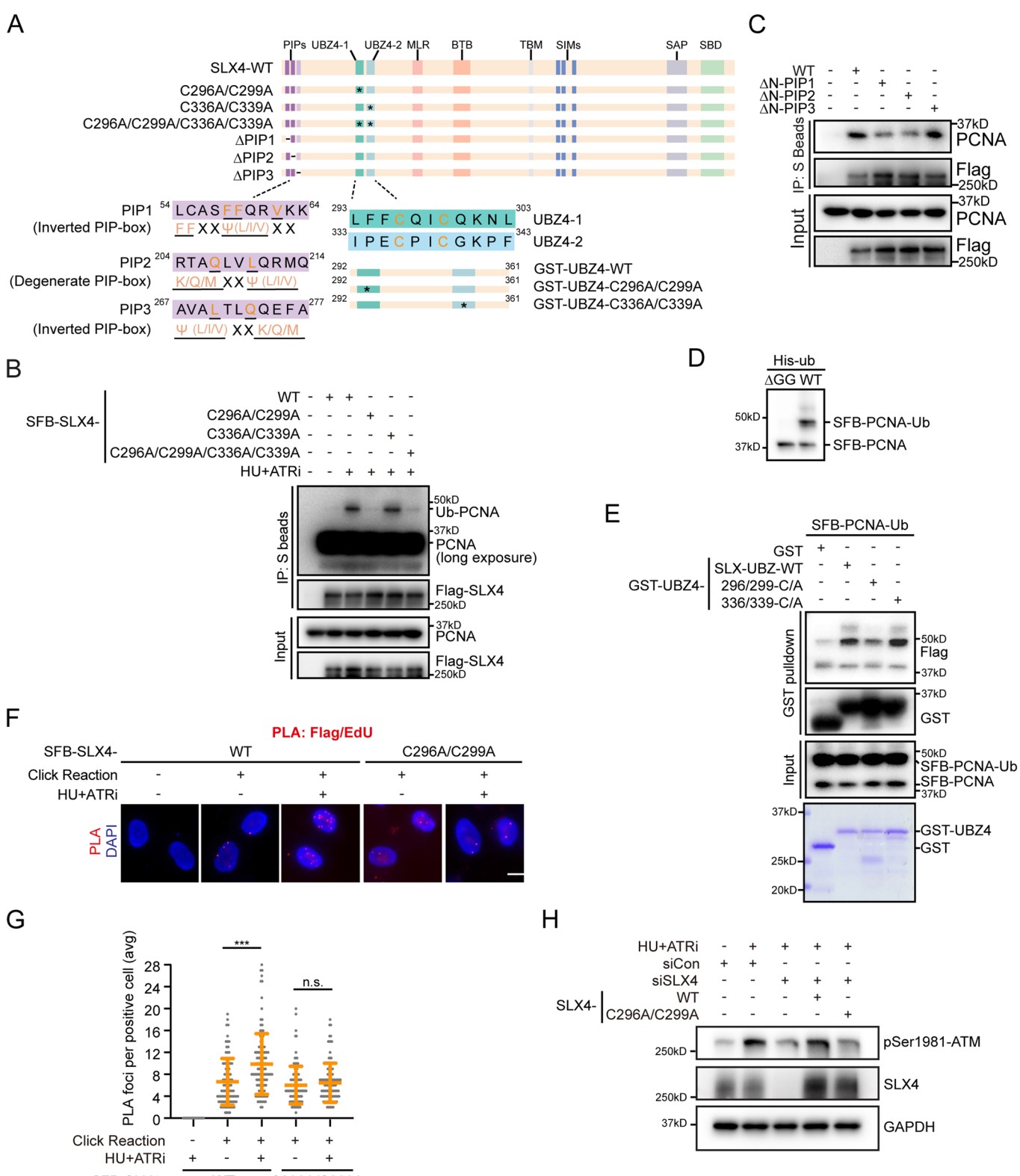

and ensuring cell viability. We demonstrate that, during replication stress, activated ATR phosphorylates human Rad18, the principal E3 ligase for PCNA, specifically at serine 403 (Fig. 7). This phosphorylation event disrupts the Rad18-PCNA interaction, subsequently restraining PCNA monoubiquitination (Fig. 7). Furthermore, our findings indicate that SLX4, a scaffolding protein for multiple endonucleases, selectively recognizes monoubiquitinated PCNA (Fig. 7). As a result, ATR-mediated human Rad18 phosphorylation functions as a safeguard, preventing excessive SLX4 accumulation at stalled forks and effectively thwarting fork collapse (Fig. 7).

Figure 5. Interaction between SLX4 and monoubiquitinated PCNA.

(A) Diagram illustrating different SLX4 mutants. (B) Interaction of SLX4 with PCNA through the UBZ4-1 domain. HEK293T cells were transfected with plasmids encoding SFB-tagged wild-type SLX4 or the indicated mutants. Twenty-four hours later, cells were mock-treated or treated with a combination of 2 mM HU and 2 μM VE-821 for 2 h. Cell lysates were subjected to immunoprecipitation with S beads and Western blot analysis was carried out as indicated. (C) HEK293T cells were transfected with plasmids encoding SFB-tagged wild-type SLX4 or the indicated mutants. 24 h later, cells were lysed in NETN buffer and subjected to immunoprecipitation with S beads. Western blot analysis was carried out as indicated. (D) Ubiquitinated SFB-tagged PCNA purified from HEK293T cells. (E) Direct binding between ubiquitinated SFB-PCNA and the UBZ4-1 domain of SLX4 in vitro. Bacterially purified GST-UBZ4 conjugated to glutathione–Sepharose resin was incubated with ubiquitinated SFB-tagged PCNA purified from HEK293T cells. Purified proteins were visualized by Coomassie staining. (F, G) UBZ4-1 domain facilitates SLX4 recruitment to stalled forks. HeLa cells transfected with plasmids encoding SFB-tagged wild-type SLX4 or the C296A/C299A mutant were mock-treated or treated with a combination of 2 mM HU and 2 μM VE-821. One hour later, cells were subjected to PLA with anti-Flag and anti-biotin antibodies. Representative images of PLA foci (red) (F). Scale bar, 10 μm. Quantification of PLA foci number per focus-positive cell (G). Data represent means ± SD of three independent experiments. More than 100 cells were counted for each sample. ***$P$ < 0.001, n.s., not significant, one-way ANOVA test. (H) HeLa cells stably expressing SFB-tagged wild-type SLX4 or the C296A/C299A mutant were generated. The resulting cells were transfected with control siRNAs or siRNAs targeting the UTRs of SLX4 as indicated. The cells were mock-treated or treated with a combination of 2 mM HU and 2 μM VE-281 for 3 h. Immunoblotting was performed using antibodies as indicated. Source data are available online for this figure.

The multifaceted SLX4–endonuclease complexes have previously been implicated in replication fork collapse (Couch et al, 2013; Fekairi et al, 2009; Forment et al, 2011; Hanada et al, 2007). SLX4 comprises a diverse array of functionally significant domains organized sequentially from the N-terminus to the C-terminus. These include two tandem UBZ4 (Ubiquitin-binding zinc finger 4) domains (UBZ4-1 and UBZ4-2), an MLR (Mus312-ME19 interaction-like) domain, a BTB (broad-complex, tramtrack, and bric à brac) domain, a TBM (TRF2-binding motif) domain, three SIMs (SUMO-interacting motifs), an SAP (SAF-A/B, Acinus, and PIAS) domain, and an SBD (SLX1-binding domain) (Guervilly and Gaillard, 2018). The presence of these diverse domains synergistically equips SLX4 with the remarkable ability to engage in a wide range of molecular interactions. For instance, SLX4 is recruited to DNA damage sites through SUMO-SIM interactions, particularly in the context of DSB repair (Guervilly et al, 2015; Ouyang et al, 2015). Additionally, SLX4 plays a pivotal role at interstrand crosslinks (ICLs) sites by interacting with ubiquitin through its UBZ4 domains (Katsuki et al, 2021; Kim et al, 2011; Lachaud et al, 2014). This interaction is facilitated by either monoubiquitinated FANCI/D2 or K63-linked polyubiquitin chains generated by the E3 ligase RNF168 (Katsuki et al, 2021). Furthermore, SLX4 localizes to telomeres in a TRF2-dependent manner through interaction mediated by the TBM domain (Wan et al, 2013; Wilson et al, 2013). However, the precise mechanisms governing SLX4 recruitment to stalled replication forks remain elusive. Our results demonstrate that SLX4 is recruited to stalled forks, at least partially, by recognizing monoubiquitinated PCNA via its PIP and UBZ4-1 motifs. Our findings provide novel insights into the diverse roles of SLX4 and its recruitment to different types of DNA lesions via distinct mechanisms, highlighting its role as a reader of monoubiquitylated PCNA.

The recruitment of Rad18 to stalled replication forks is believed to involve RPA-coated ssDNA (Davies et al, 2008). However, the mechanism underlying the interaction between Rad18 and PCNA has remained elusive. In our study, we have identified a non-classical PIP motif on human Rad18 that facilitates the interaction between Rad18 and PCNA. Importantly, we have demonstrated that this interaction can be modulated by ATR. We propose that phosphorylation of human Rad18, particularly at serine 403, could induce conformational changes in the adjacent PIP motif, thereby influencing its interaction with PCNA. Nonetheless, the precise mechanism through which serine 403 phosphorylation disrupts the

Rad18-PCNA interaction and the intricate details of the Rad18-PCNA complex demand further investigation, particularly through structural studies.

Interestingly, the PIP motif within Rad18 displays a remarkable degree of conservation across diverse species. However, a distinct pattern emerges, where the serine residue at position 403 is notably exclusive to higher primates. In contrast, in other evolutionary lineages, this specific position is conventionally occupied by a proline residue. Aligning with this evolutionary divergence, it's worth noting that ATR in murine cellular contexts doesn't significantly impede PCNA monoubiquitination. This intriguing observation suggests that ATR's regulatory mechanism, aimed at restraining PCNA monoubiquitination to prevent excessive SLX4 accumulation at stalled replication forks, might have undergone gradual refinement. This adaptation likely corresponds to the increased genomic intricacies characteristic of more advanced biological organisms.

Previous studies have noted the inhibitory effect of ATR inhibition on ALT activity, although the exact mechanism remains elusive (Flynn et al, 2015). Additionally, studies have also indicated that overexpression of SLX4 in ALT cells interferes with the ALT pathway (Sobinoff et al, 2017). In our study, we observed that inhibiting ATR activity in G2 phase U2OS cells not only enhances the monoubiquitination of PCNA but also triggers a concurrent increase in the accumulation of SLX4 at telomeres in ALT cells. Intriguingly, the inhibitory effect of ATR inhibition on ALT is contingent upon SLX4. Furthermore, upon depleting Rad18, the increased telomeric SLX4 accumulation caused by ATR inhibition, as well as the inhibitory impact on ALT, were both restored, underscoring ATR's functional role in ALT protection through the Rad18 signaling pathway. However, considering that excessive SLX4 itself can trigger the collapse of stalled replication forks and subsequent DNA breakage, potentially yielding opposing effects by facilitating break-induced replication (BIR) (Dilley et al, 2016; Roumelioti et al, 2016; Zhang et al, 2019), selecting appropriate ATR concentrations for treatment in different ALT cancer cell lines might be a critical consideration. Notably, a recent study revealed that the interaction orchestrated by Rad18 between ubiquitinated PCNA and the SNM1A nuclease plays a crucial role in the formation of recombination intermediates at damaged telomeres (Zhang et al, 2023). While SNM1A localization to damaged telomeres largely depends on Rad18, Rad18 depletion only prevents excessive SLX4 accumulation at telomeres caused by ATR inhibition, indicating that SLX4 accumulation at telomeres can also be regulated by other proteins such as TRF2. The potential regulation of SNM1A recruitment to damaged telomeres by the ATR kinase remains an area for future investigation.

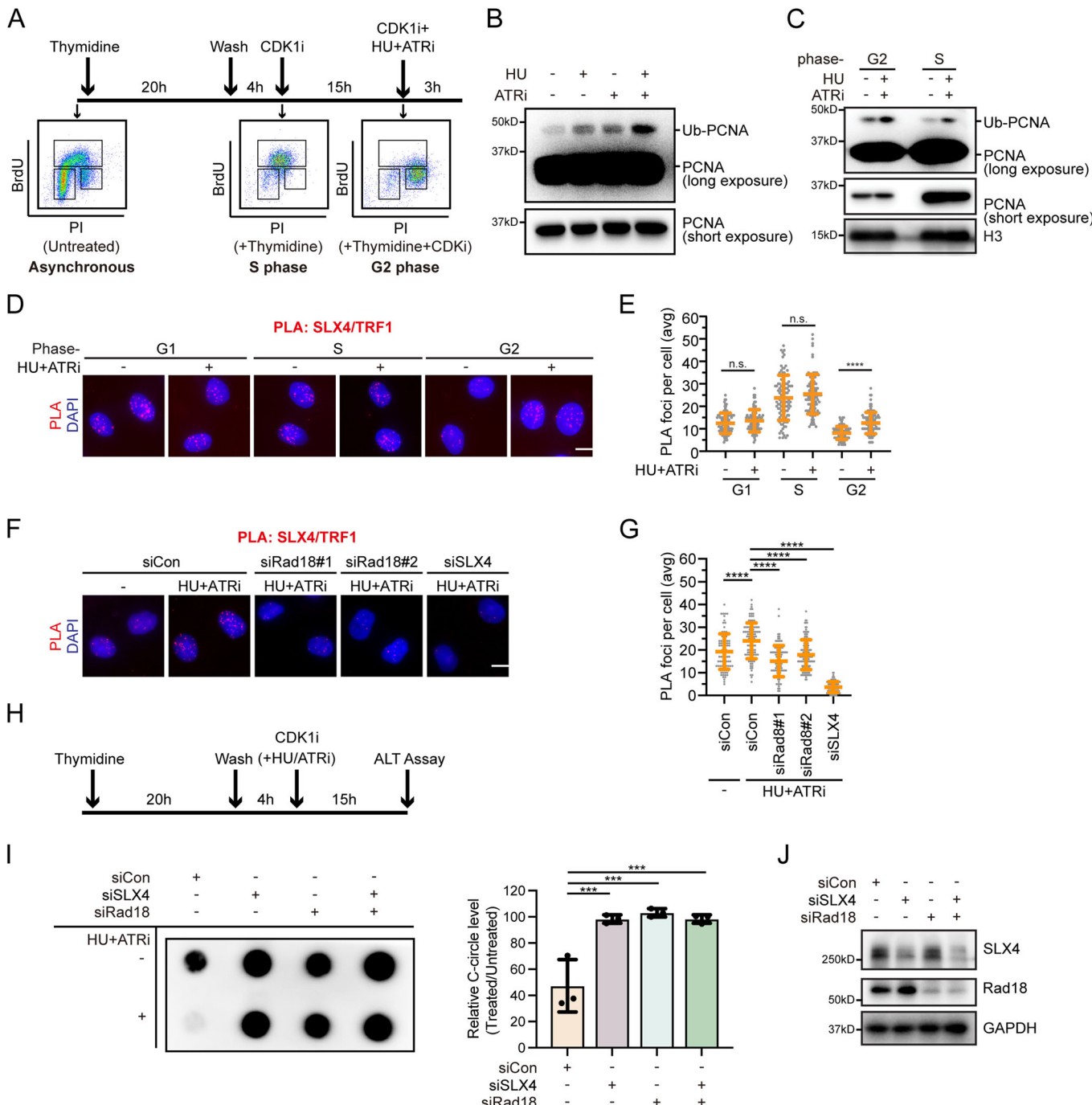

## Methods

### Antibodies

Rabbit polyclonal anti-Rad18 (WB dilution: 1:500), anti-SLX4 (WB dilution: 1:100, immunostaining dilution: 1:20,000), and anti-XPF (WB dilution: 1:1000) were produced by immunizing rabbits with GST-Rad18 (amino acids 295–495), GST-SLX4 (amino acids 500-900), and MBP-XPF (amino acids 616–916) purified from *E. coli* (Hangzhou HuaAn Biotechnology). Anti-phospho-Chk1 (Ser296) (ET1611-76, WB dilution: 1:500) was purchased from Hangzhou HuaAn Biotechnology. Anti-

phospho-Chk1 (Ser345) (2348, WB dilution: 1:500), anti-phospho-Rad18 (Ser403) (14978, WB dilution: 1:1000), and anti-Ubiquityl-PCNA (Lys164) (13439, WB dilution: 1:1000) were purchased from Cell Signaling Technology. Anti-phospho-ATM (ab81292, WB dilution: 1:10,000), anti-phospho-MCM2 (ab133243, WB dilution:1:1000), anti-MBP (ab119994, WB dilution: 1:2,000), anti-TRF1 (ab10579, immunostaining dilution: 1:10,000) and anti-CldU/BrdU (ab6326, immunostaining dilution: 1:400) antibodies were purchased from Abcam. Anti-IdU/BrdU (B44) (347580, immunostaining dilution: 1:400) antibody and anti-BrdU for BrdU incorporation assay (55627, immunostaining dilution: 1:200) was purchased from BD Biosciences. Anti-BrdU for native

**Figure 6. ATR inhibits excessive SLX4 accumulation at telomeres in ALT cells.**

(A) Top: Schematic analysis of PCNA monoubiquitination in U2OS cells enriched in the S phase or G2 phase. For U2OS cells enriched in the S phase, cells were treated with thymidine for 20 h, released into DMEM for 4 h, and collected. For U2OS cells enriched in the G2 phase, cells were treated with thymidine for 20 h, released into DMEM for 4 h, and then treated with CDK1 inhibitor, 15 μM RO-3306, for 15 h. Bottom: BrdU incorporation and cell-cycle profiles of the tested cell populations. (B) ATR inhibits PCNA monoubiquitination in U2OS cells during the G2 Phase. U2OS cells synchronized in G2 were treated with a combination of 2 mM HU and 2 μM VE-821 or an equivalent volume of DMSO for 3 h. Chromatin fractions were isolated and immunoblotted with the indicated antibodies. (C) U2OS cells synchronized in the G2 phase or S phase were treated with a combination of 2 mM HU and 2 μM VE-821 or an equivalent volume of DMSO for 3 h. Chromatin fractions were isolated and immunoblotted with the indicated antibodies. (D, E) ATR limits excessive SLX4 accumulation at telomeres in U2OS Cells. For G1 phase enrichment, cells were treated with thymidine for 20 h. For S phase enrichment, cells were treated with thymidine for 20 h, and released into DMEM for 2 h. For G2 phase enrichment, cells were treated with thymidine for 20 h, released into DMEM for 4 h, and then treated with CDK1 inhibitor, 15 μM RO-3306, for 15 h. Subsequently, U2OS cells enriched in each phase were either mock-treated or treated with 2 mM HU and 2 μM VE-821 for 3 h, followed by PLA using anti-SLX4 and anti-TRF1 antibodies. Representative images of PLA foci (D). Scale bar, 10 μm. Quantification of PLA foci number per focus-positive cell (E). Data represent means ± SD of three independent experiments. More than 100 cells were counted for each sample. n.s. indicates not significant, ****$P < 0.0001$, one-way ANOVA test. (F, G) U2OS cells were transfected with the indicated siRNAs. Twenty-four hours after transfection, cells were synchronized in the G2 phase, mock-treated or treated with 2 mM HU and 2 μM VE-821 for 3 h, and then subjected to PLA using anti-SLX4 and anti-TRF1 antibodies. Representative images of PLA foci (F). Scale bar, 10 μm. Quantification of PLA foci number per focus-positive cell (G). Data represent means ± SD of three independent experiments. More than 100 cells were counted for each sample. ****$P < 0.0001$, one-way ANOVA test. (H) Schematic analysis of the effects of ATR inhibition on ALT activity. U2OS cells were treated with thymidine for 20 h, released into DMEM for 4 h, and then treated with CDK1 inhibitor RO-3306 (15 μM) in combination with HU (50 μM) and VE-821 (500 nM). After 15 h of drug treatment, cells were collected for the C-circle assay. (I) Left: A representative dot blot for telomeric C-circle assay showing levels of telomeric C-circles in U2OS cells, either mock-treated or treated with HU+ATRi. Right: C-circle levels are quantified as percentages relative to those in mock-treated cells. Data represent means ± SD of three independent experiments. ***$P < 0.001$, one-way ANOVA test. (J) Western blot analysis of knockdown efficiency of Rad18 and SLX4. Source data are available online for this figure.

immunostaining (RPN202; immunostaining dilution: 1:1000) was purchased from GE Healthcare. Anti-phospho-CDK1 (AP0016, WB dilution: 1:1,000) and anti-MUS81 (A6818, WB dilution: 1:500) were purchased from ABclonal Technology. Anti-SLX1 (21158-1-AP, WB dilution: 1:500) were purchased from Proteintech. Anti-H3 (04-928, WB dilution: 1:5000) and anti-GAPDH (MAB374, WB dilution: 1:1000) were purchased from EMD Millipore. Anti-PCNA (PC10) (sc-56, WB dilution: 1:1000) antibodies were purchased from Santa Cruz Biotechnology. Anti-RPA1 (NA13, WB dilution: 1:1000) was purchased from Calbiochem. Anti-Flag (M2, WB dilution: 1:5000, immunostaining dilution: 1:10,000) was purchased from Sigma-Aldrich. Rabbit anti-Biotin (150-109 A, immunostaining dilution: 1:3000) antibodies were purchased from Bethyl. Mouse anti-biotin (200-002-211, immunostaining dilution: 1:4000) antibodies, Rhodamine-conjugated goat anti-mouse IgG (15-001-003, immunostaining dilution: 1:500), FITC-conjugated goat anti-mouse IgG (115-095-146, immunostaining dilution: 1:200), and FITC-conjugated goat anti-rabbit IgG (111-095-003, immunostaining dilution: 1:100) were purchased from Jackson ImmunoResearch. Alexan Fluor 488-conjugated donkey anti-rat IgG (A-21208, immunostaining dilution: 1:400) was purchased from Life Technologies.

## Inhibitors

The ATR inhibitor VE-821 (S8007), Chk1 inhibitor MK-8776 (S2735), Wee1 inhibitor MK-1775 (S1525), CDC7 inhibitor PHA-767491 (S2742), and CDK1 inhibitor RO-3306 (S7747) were purchased from Selleck Chemicals and solubilized in dimethylsulfoxide (DMSO). Chk1 inhibitor UCN-01 (U6508) was purchased from Sigma-Aldrich. These inhibitors were stored at −80 °C and employed in accordance with the manufacturer's guidelines.

## RNAi

All siRNAs used in this study were chemically synthesized by RuiBo. The sequences of siRNAs are as follows: Rad18 siRNA#1: 5′-GCCGGAUCUGAAAAAUAAC-3′; Rad18 siRNA#2: 5′-CCAGCC AAAUCUCCUGCUUTT-3′; ATR siRNA #1: 5′-AACCUCCGUGA UGUUGCUUGA-3′; ATR siRNA #2: 5′-AAGCCAAGACAAAU

UCUGUGU-3′; PCNA siRNA: 5′-GCCGAGAUC UCAGCCAU AUTT-3′; SLX4 siRNA #1: 5′-GGAACGAAGTCGCACAGAA-3′; SLX4 siRNA#2: a mixture of siRNAs targeting the 5′-UTR (5′-GC ACCAGGUUCAUAUGUAUTT-3′) and 3′-UTR (5′-GCACAAG GGCCCAGAACAATT-3′); MUS81 siRNA#1: 5′-CAUUAAGUGU.

GGGCGUCUA-3′; MUS81 siRNA#2: 5′-GAGUUGGUACUGG AUCACAUU-3′;XPF siRNA#1: 5′-GUAGGAUACUUGUGGUU GA-3′; XPF#2: 5′-ACAAGACAAUCCGCCAUUA-3′; SLX1 siRNA#1:5′-UGGACAGACCUGCUGGAGA-3′;
SLX1 siRNA#2:5′- CCTGGCAGAGGAGTTT.

CTT-3′; SLX1 siRNA#3:5′-CCAGATGGACACTGAGAAA-3′; and control siRNA: 5′-UUCAA UAAAUUCUUGAGGUUU-3′. To generate siRNA-resistant Rad18 plasmids, six nucleotides in the Rad18 siRNA#1-targeting region were substituted (G210A, T213C, G215A, A218G, T221C, and C224T). For siRNA transfection, cells were transfected twice at a 24-h interval with the indicated siRNAs using Lipofectamine RNAiMAX (Invitrogen) following the manufacturer's protocol.

## Cell culture

Human HeLa and HEK293T cells were obtained from ATCC and cultured in Dulbecco's modified Eagle's medium (DMEM; GIBCO C11995500BT) supplemented with 10% fetal bovine serum (GIBCO 10270-106) and 1% penicillin and streptomycin (Hyclone SV30010)), and maintained at 37 °C with 5% CO$_2$. Cell lines were routinely tested for mycoplasma contamination and maintained in cultures for no more than 1 month.

## Plasmids and transfection

The cDNA sequences encoding full-length human PCNA, Rad18, or SLX4 were amplified through PCR and subsequently cloned into pDONOR201 or pDONOR221 vectors utilizing the Gateway Technology (Invitrogen). The corresponding fragments within the entry clone were subsequently recombined into a gateway-compatible destination vector, which carries either an N-terminal SFB-tag (S tag, Flag epitope tag, and streptavidin-binding peptide tag) or an

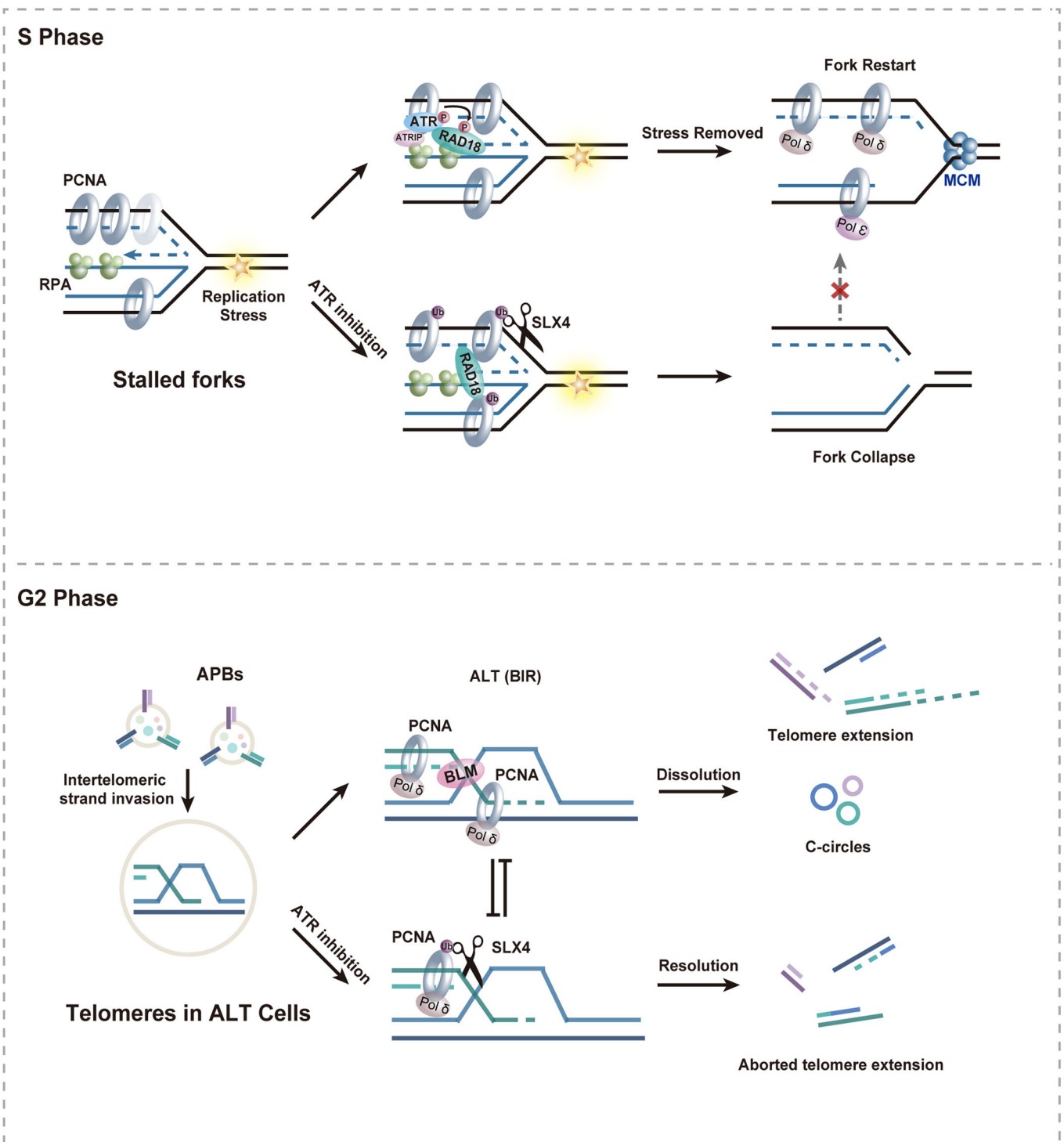

**Figure 7.   Working model depicting a proposed role of ATR in preserving replication fork and telomerase-independent telomere stability.**

Upon replication stress, ATR phosphorylates human Rad18 at Serine 403, disrupting its interaction with PCNA. This action restricts the extent of Rad18-mediated PCNA monoubiquitination. Consequently, excessive accumulation of the tumor suppressor protein SLX4 at stalled forks and ALT telomeres is prevented, thus maintaining replication fork and telomerase-independent telomere stability.

N-terminal HF-tag (HA tag, Flag tag). Site-directed mutagenesis was carried out using PCR-mediated methods in accordance with standard protocols to generate mutants of PCNA, Rad18, or SLX4. For achieving doxycycline-inducible protein expression, the cDNA encoding wild-type or mutant variants of PCNA was integrated into a doxycycline-inducible lentiviral vector featuring a Tet-On–inducible C-terminal SFB-tag via the utilization of the Gateway technology. All plasmids utilized in this study were subjected to sequencing verification. For transfections in HEK293T cells, plasmids were introduced using polyethyleneimine (Yeasen Biotechnology) following the manufacturer's instructions. In the case of transfections in HeLa cells, plasmids were introduced using Hieff Trans™ Liposomal Transfection Reagent (Yeasen Biotechnology) following the manufacturer's guidelines. For transfections with both siRNAs and plasmids, cells were transfected with the indicated siRNAs using Lipofectamine RNAiMAX transfection reagent (Invitrogen) following the manufacturer's instructions. Forty-eight hours after siRNA transfection, cells were repassaged and subsequently transfected with the indicated plasmids using Hieff Trans™ Liposomal Transfection Reagent following the manufacturer's guidelines (Yeasen Biotechnology).

## Retrovirus production and infection

Plasmids containing Rad18 from pDONR201 derivatives were transferred into a Gateway-compatible retroviral destination vector, pEF1A-HA-Flag. Retroviruses were generated in HEK293T cells through co-transfection of the retrovirus-expressing plasmid with the pCL-ECO packaging plasmid and the VSV-G envelope plasmid. After 48 h of transfection, retroviral supernatants were collected and used to infect HeLa cells in the presence of 8 µg/mL Polybrene (Sigma, H9268). Stable cell lines were established by selecting with medium containing 2 µg/mL puromycin (Merck, 540411).

## Lentivirus packaging and infection

The Tet-On–inducible SFB-tagged lentiviral vector and packaging plasmids (pMD2G and pSPAX2) were generously provided by S. Zhou (Baylor College of Medicine, Houston, TX). Virus supernatant was harvested 48 h after the co-transfection of lentiviral vectors and packaging plasmids (pMD2G and pSPAX2) into HEK293T cells. HeLa cells were subjected to viral supernatant infection with the addition of 8 µg/ml polybrene (Sigma-Aldrich) and subsequently selected in a growth medium containing 500 µg/ml G418 (EMD Millipore).

## Chromatin fractionation

Chromatin fractionation was conducted following a modified protocol described previously (Huang et al, 2009). Briefly, cells were collected after drug treatment and washed with PBS. Cell pellets were then suspended in NETN buffer (20 mM Tris-HCl [pH 8.0], 100 mM NaCl, 1 mM EDTA, and 0.5% Nonidet P-40) containing protease inhibitors (1 µg/ml aprotinin and leupeptin) and incubated on ice for 15 min. Nuclei were subsequently isolated and resuspended in 0.2 M HCl. The soluble fraction was then neutralized with 1 M Tris-HCl, pH 8.0.

## Co-immunoprecipitation and western blotting

To assess the interaction between ubiquitinated PCNA and SLX4, HEK293T cells were transfected with SFB-tagged SLX4.

Subsequently, cells were treated with 2 mM HU and 2 µM VE-821 for 2 h before cross-linking with 1% formaldehyde for 15 min. To quench cross-linking, glycine was added to a final concentration of 0.125 M for 5 min. The collected cell pellets were resuspended in CSK buffer (10 mM PIPES [pH 6.8], 300 mM sucrose, 100 mM NaCl, 1 mM EGTA, 3 mM magnesium chloride, and 0.5% [v/v] Triton X-100) containing protease inhibitors (1 µg/ml aprotinin and leupeptin) and incubated for 5 min at 4 °C. Lysates were then centrifuged at $1300 \times g$ for 5 min, and the supernatant was removed as the soluble fraction. Nuclei were washed once with CSK buffer, lysed in NETN buffer containing protease inhibitors (1 µg/ml aprotinin and leupeptin), and subjected to sonication. The resulting crude lysates were centrifuged at $14{,}000 \times g$ for 6 min, and the supernatants were collected and incubated with S protein beads at 4 °C for 2 h on a rotator. The Sepharose beads were subsequently washed with NTEN buffer three times, boiled in 2× SDS loading buffer, and separated on SDS-PAGE. Membranes were blocked with 5% milk in TBST buffer (0.2 M Tris-HCl [pH 8.0], 0.6 M NaCl, 0.8% Tween 20), followed by probing with the specified antibodies.

## Recombinant protein purification

For GST-PCNA and GST-SLX4-UBZ proteins, the coding sequences of PCNA or SLX4-UBZ4 wild-type and mutant fragments were cloned into pDONR201 as entry clones and subsequently transferred to the pDEST15 destination vector (Invitrogen) for GST-tagged fusion protein expression in E. coli. Bacterial cells were cultured at 37 °C to mid-log phase and induced with 0.2 mM isopropyl β-D-1-thiogalactopyranoside (IPTG) at 16 °C for 16 h. After harvesting, cells were resuspended in lysis buffer (20 mM Tris-HCl [pH 8.0], 300 mM NaCl, 1% Triton X-100, 1 mM EDTA, 2 mM DTT, 1 µg/ml aprotinin and leupeptin), followed by sonication. Clarified lysates were then incubated with glutathione–Sepharose resin (Thermo Scientific) at 4 °C for 1 h. The beads were centrifuged, washed with a washing buffer (20 mM Tris-HCl [pH 8.0], 500 mM NaCl, 0.5% NP-40, 2 mM DTT) containing protease inhibitors, and used for pull-down assays or eluted with 20 mM reduced glutathione for in vitro pull-down assays.

For MBP-PCNA and MBP-Rad51 proteins, the coding sequences of PCNA or Rad51 were cloned into pDONR201 as entry clones, and subsequently transferred to pDEST15 destination vector (Invitrogen) for MBP-tagged fusion protein expression in E. coli. After harvesting and resuspension in lysis buffer, followed by sonication, clarified lysates were incubated with amylose resins at 4 °C for 1 h. The beads were centrifuged, washed with a washing buffer (20 mM Tris-HCl [pH 8.0], 500 mM NaCl, 0.5% NP-40, 2 mM DTT) containing protease inhibitors, and used for pull-down assays.

For His-sumo-Rad18 proteins, full-length Rad18 was cloned into the pET28-N-His-sumo vector (EMD Millipore) for His-sumo-tagged Rad18 expression in E. coli (the sumo tag on RAD18 was able to improve its solubility). Harvested cells were resuspended in lysis buffer without EDTA (20 mM Hepes, 300 mM NaCl, 1% Triton X-100, 2 mM DTT, 1 µg/ml aprotinin and leupeptin), followed by sonication. Clarified lysates were then incubated with cobalt agarose at 4 °C for 2 h. After washing the beads with a washing buffer (20 mM Hepes, 500 mM NaCl, 1%

Triton X-100, 5 mM imidazole, 1 µg/ml aprotinin and leupeptin), the bound proteins were eluted using lysis buffer containing 200 mM imidazole.

## Preparation of ubiquitinated PCNA

HEK293T cells stably expressing His-tagged wild-type Ub or Ub-ΔGG were transfected with plasmids encoding SFB-tagged PCNA. After 24 h of transfection, cells were treated with 2 mM HU and 2 mM ATRi for 2 h. Subsequently, cells were collected, lysed, and sonicated in denaturing Buffer A (6 M guanidine–HCl, 100 mM NaH2PO4/Na2HPO4, and 15 mM imidazole). Clarified lysates were incubated with cobalt resin (Thermo Scientific) at 4 °C overnight. The beads were then centrifuged and washed once with Buffer A, once with Buffer B (25 mM Tris-HCl [pH 6.8] and 10 mM imidazole), and the bound ubiquitinated proteins were eluted using NETN buffer containing 200 mM imidazole for 1 h. The eluent was incubated with streptavidin-conjugated beads (GE Healthcare) for 1 h at 4 °C. The beads were washed with NETN buffer containing 300 mM NaCl three times, and then the ubiquitinated SFB-tagged PCNA was eluent with NETN buffer containing 1 mg/mL biotin (Sigma-Aldrich).

## In vitro pull-down assay

The pull-down assay using GST-tagged proteins was conducted in NETN buffer at 4 °C for 1 h on a rotator. Subsequently, the beads were subjected to three washes with NTEN buffer. For the pull-down assay involving His-sumo-tagged Rad18 bound to cobalt agarose, the procedure was performed in NETN buffer without EDTA, containing 5 mM imidazole and 300 mM NaCl at 4 °C for 45 min on a rotator. The samples were boiled in 2× SDS loading buffer and separated using SDS-PAGE.

## Proximity ligation assay

To detect PLA signals at stalled forks, HeLa cells in exponential growth were pulse-labeled with 10 µM EdU for 15 min, followed by treatment with 2 mM HU with or without 2 µM VE-821 for 1 h. For detecting PLA signals at telomeres in ALT cells, U2OS cells were initially treated with 2 mM thymidine for 20 h, released into fresh medium for 4 h, and subsequently treated with 15 µM CDK1 inhibitor RO-3306 for 15 h. The addition of 2 mM HU with 2 µM VE-821 to the medium, along with RO-3306, was carried out for an additional 3 h. Cells were then washed with PBS, pre-extracted in CSK buffer for 5 min, fixed with 3% (w/v) paraformaldehyde (PFA, pH 7.0) for 10 min, and permeabilized with PBS containing 0.5% (v/v) Triton X-100 for 5 min. After PBS washes, cells underwent the Click-iT reaction to conjugate biotin to EdU. Subsequently, the cells were blocked with 3% (w/v) BSA in PBS for 30 min. Following blocking, cells were subjected to overnight incubation with the relevant primary antibodies at 4 °C. The proximity ligation assay was conducted using the Duolink In Situ Red Starter kit (Sigma-Aldrich), following the manufacturer's instructions. Image acquisition was performed using a Nikon Eclipse 80i Fluorescence Microscope equipped with a Plan Fluor 60× oil objective lens (NA 0.5–1.25; Nikon) and a camera (CoolSNAP HQ2; Photometrics), with subsequent analysis using NIS-Elements basic research imaging software (Nikon). The data presented are representative of three independent experiments.

## Detection of nascent single-stranded DNA (ssDNA) by native BrdU assay

To detect nascent ssDNA, HeLa cells were subjected to a pulse-labeling with 10 µM BrdU (Sigma-Aldrich) for 20 min prior to treatment with 2 mM HU and 2 µM VE-821 for 3 h. Following a PBS wash, cells were permeabilized with 0.5% Triton X-100 for 5 min at 4 °C and subsequently fixed with 3% paraformaldehyde for 10 min at room temperature, without undergoing any DNA denaturation treatment. Fixed cells were then incubated with mouse anti-BrdU antibody (GE, RPN202, 1:1000) for 20 min at room temperature, followed by Rhodamine-conjugated goat anti-mouse IgG. To visualize nuclear DNA, counterstaining was performed with DAPI. Image capture was conducted using a Nikon Eclipse 80i Fluorescence Microscope equipped with a Plan Fluor 60× oil objective lens (NA 0.5–1.25; Nikon) and a camera (CoolSNAP HQ2; Photometrics), and subsequent analysis was carried out utilizing NIS-Elements basic research imaging software (Nikon).

## DNA fiber analysis

HeLa cells were subjected to pulse-labeling with 50 µM IdU (Sigma-Aldrich, I7125) for 30 min, followed by treatment with 2 mM HU (and 2 µM VE-821) for 1 h, and then another pulse-labeling with 100 µM CldU (Sigma-Aldrich, C6891) for 30 min. The labeled cells were rapidly trypsinized and suspended in ice-cold PBS at a concentration of $1 \times 10^6$ cells/ml. Subsequently, 2.5 µl of the cell suspension was spotted onto on a pre-cleaned glass slide and lysed using 7.5 µl of spreading buffer (0.5% SDS, 50 mM EDTA, 200 mM Tris-HCl [pH 7.4]). After 5 min, the slides were tilted at a 15° angle relative to the horizontal, leading to the formation of DNA spreads. These spreads were air-dried, fixed in a 3:1 methanol/acetic acid solution for 20 min, and then denatured with 3 M HCl overnight at 4 °C. Following a PBS wash, the slides were blocked with 1% BSA in PBS for 30 min at room temperature and subsequently incubated with anti-IdU/BrdU (BD Biosciences, clone B44, 1:400) and anti-CldU/BrdU (Abcam, ab2326, 1:400) antibodies for IdU and CldU detection, respectively. After a 3-h incubation, the slides were washed with PBS and stained with Rhodamine-conjugated goat anti-mouse IgG (Jackson Immunoresearch Laboratories, 1:400) and Alexa Fluor 488 Donkey anti-Rat IgG (Life Technologies, 1:400) for 2 h at room temperature in the dark. Images were captured using a Nikon Eclipse 80i Fluorescence Microscope equipped with a Plan Fluor 60 × oil objective lens (NA 0.5–1.25; Nikon), and a camera (CoolSNAP HQ2; Photometrics), and were analyzed using NIS-Elements basic research imaging software (Nikon). The presented data represent a minimum of three independent experiments, with the measurement of at least 100 individual tracks in each experiment.

## Cell survival assay

HeLa cells transfected with the indicated siRNAs were seeded into 12-well plates at a density of $2 \times 10^2$ cells per well and allowed to incubate for 18 h. Subsequently, the cells were treated with HU and VE-821 at the specified concentrations for a duration of 10 days. After the incubation period, the formed colonies were fixed and stained with Coomassie blue.

## BrdU incorporation assays

U2OS cells were initially treated with 2 mM thymidine for 20 h, released into fresh medium for 4 h, and subsequently exposed to a 15 μM CDK1 inhibitor RO-3306 for 15 h. Following this, 2 mM HU was added along with or without 2 μM VE-821 for an additional 3 h in the presence of the CDK1 inhibitor. Cells were then fixed using ice-cold 70% ethanol after a 1-h incubation with BrdU (100 mM). Upon centrifugation, cells were washed with PBS. DNA denaturation was achieved by treating with 2.5 M HCl for 1 h at room temperature. After triple washing with PBS, cells were subjected to a 12-h incubation with an anti-BrdU antibody, followed by a three-time wash with a blocking buffer containing 500 mM NaCl. Subsequently, FITC-conjugated goat anti-mouse IgG was added and allowed to incubate for 4 h. Following another wash with a blocking buffer containing 500 mM NaCl, cells were resuspended with propidium iodide (20 mg/mL) and RNase A (200 mg/mL). The distribution of the cell cycle was analyzed using a flow cytometer (FACScan; Beckman Coulter).

## VE-821 toxicity analysis in a panel of 1001 cancer cell lines

Toxicity analysis of VE-821 in a panel of 1001 cancer cell lines was conducted using IC50 data retrieved from the Genomics of Drug Sensitivity in Cancer database (https://www.cancerrxgene.org/). The pharmacogenomics dataset, containing information on 1001 human cancer cell lines, was obtained from the ArrayExpress database under accession code E-MTAB-3610 (Iorio et al, 2016). Subsequently, we employed the Pearson correlation coefficient to evaluate the relationship between gene expression levels and IC50 values of VE-821.

## C-circle assay

U2OS cells were treated with 2 mM thymidine for 20 h, released into fresh medium for 4 h, and subsequently treated with 15 μM RO-3306, along with 50 μM HU and 500 nM VE-821 for 15 h. Genomic DNA was extracted using sodium dodecyl sulfate (SDS) lysis buffer (2% SDS, 50 mM Tris, 20 mM ethylenediaminetetraacetic acid (EDTA)) containing 200 μg/ml Pronase protease (Sigma) at 60 °C for 2 h, followed by addition of 5 M sodium chloride. After centrifugation for 30 min at 15,000 × g, the supernatant containing genomic DNA was precipitated with ethanol at −20 °C overnight. The samples were then centrifuged for 30 min at 15,000 × g at 4 °C. The resulting pellets were washed once with 70% (v/v) ethanol, air-dried, and suspended in 10 mM Tris (pH 7.6). DNA was quantified using a Nanodrop spectrophotometer. Diluted DNA (120 ng) was mixed with a 10 μl reaction buffer containing 0.2 mg/ml BSA (NEB), 0.1% Tween, 4 mM dithiothreitol (DTT), 1 mM of each dNTP except dCTP (Thermo Fisher), 1× Φ29 Buffer (NEB), and 7.5 U Φ29 DNA polymerase (NEB). Samples were incubated at 30 °C for 8 h, followed by 65 °C for 20 min. The resulting C-circle amplification DNA products were diluted with 100 μl of 2× SSC buffer and dot-blotted onto a nylon membrane soaked in 2 × SSC. The amplification DNA products were then UV-cross-linked onto the membrane and detected using the Telo TAGGG telomere length kit (Roche) according to the manufacturer's instructions. Quantification of C-circle products was performed using ImageJ.

## Quantification and statistical analysis

All experiments reported were independently replicated at least three times with similar results. Data were mean ± SD from at least three independent experiments. The statistical significance of data were determined using one-way ANOVA with Dunnett's post hoc test or $t$-test, performed using GraphPad Prism Version 9. Significance is indicated by asterisk ($^{****}P < 0.0001$; $^{***}P < 0.001$; $^{**}P < 0.01$; $^{*}P < 0.05$; n.s., not significant) and $P < 0.05$ was considered statistically significant.

## Data availability

All data have been included in the manuscript, Figures, and the Source Data files. This study includes no data deposited in external repositories.

## Peer review information

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

## Acknowledgements

We thank all members of the Huang and Liu groups for insightful discussions. This work was supported by the National Key Research and Development Program of China [2022YFA1302800 and 2021YFA1101000] and the National Natural Science Foundation of China [31961160725, 31730021, 31971220, 32270769, 31970664, and 31822031].

## Author contributions

**Siyuan Chen**: Data curation; Formal analysis; Validation; Investigation; Visualization; Methodology. **Chen Pan**: Formal analysis. **Jun Huang**: Conceptualization; Supervision; Funding acquisition; Writing—original draft; Writing—review and editing. **Ting Liu**: Conceptualization; Supervision; Funding acquisition; Writing—original draft; Writing—review and editing.

## Disclosure and competing interests statement

The authors declare no competing interests.

# Expanded View Figures

**Figure EV1.   ATR inactivation hampered the restart of stalled forks.**

(**A**) Rad18 depletion improves fork restart efficiency in the absence of ATR under replication stress. Top: Schematic of the DNA fiber experiment. HeLa cells were transfected with the indicated siRNAs. Forty-eight hours after transfection, cells were incubated with 50 μM IdU for 30 min, treated with 2 mM HU with or without 2 μM VE-821 for 1 h, and incubated with 100 μM CldU for 30 min. Bottom: representative IdU and CldU replication tracks in cells transfected with indicated siRNAs. (**B**) Dot plot of CldU to IdU track length ratios for individual replication forks. Data represent means ± SD of three independent experiments. More than 100 fibers were analysed for each sample. n.s. indicates not significant, ****$P < 0.0001$, one-way ANOVA test. (**C**)Short-term ATR inhibition does not affect fork degradation under replication stress. Top: schematic of the DNA fiber experiment. HeLa cells were incubated with 50 μM IdU for 30 min, washed with PBS three times, and incubated with 100 μM CldU for 30 min. Then, cells were washed with PBS three times and treated with 2 mM HU with or without 2 μM VE-821 for 1 h. Bottom: representative IdU and CldU replication tracks. (**D**) Dot plot of CldU to IdU track length ratios for individual replication forks. Data represent means ± SD of three independent experiments. More than 100 fibers were analysed for each sample. n.s. indicates not significant, one-way ANOVA test. (**E**) Simultaneous depletion of Rad18 and SLX4 did not induce a further increase in fork restart efficiency. Top: Schematic of the DNA fiber experiment. HeLa cells were transfected with the indicated siRNAs. Forty-eight hours after transfection, cells were incubated with 50 μM IdU for 30 min, treated with 2 mM HU and 2 μM VE-821 for 1 h, and incubated with 100 μM CldU for 30 min. Bottom: representative IdU and CldU replication tracks in cells transfected with indicated siRNAs. (**F**) Dot plot of CldU to IdU track length ratios for individual replication forks. Data represent means ± SD of three independent experiments. More than 100 fibers were analysed for each sample. n.s. indicates not significant, ****$P < 0.0001$, one-way ANOVA test. (**G**) ZRANB3-mediated fork reversal is not responsible for fork collapse in the absence of ATR under replication stress. HeLa cells were transfected with the indicated siRNAs. After 48 h of transfection, cells were treated with 2 mM HU and 2 μM VE-821 for 3 h. Cell lysates were then prepared, and western blot analysis was carried out as indicated. (**H**) ZRANB3 is not required for nascent-strand ssDNA generation at stalled forks when ATR is inhibited. Top: Schematic of the native BrdU immunofluorescence assay for nascent ssDNA detection. HeLa cells were transfected with the indicated siRNAs. After 48 h of transfection, cells were labeled with 10 μM BrdU for 15 min and then either mock-treated or treated with 2 mM HU and 2 μM VE-821. After 3 h, cells were fixed and stained with an antibody against BrdU without DNA denaturation to selectively detect nascent-strand ssDNA. Bottom: Representative BrdU foci in cells transfected with the indicated siRNAs. Scale bar, 10 μm. (**I**) Quantification of BrdU foci. Cells with more than five BrdU foci were considered positive. Each black dot in the graph represents the percentage of BrdU-positive cells in each measurement. Data represent means ± SD of three independent experiments. More than 200 cells were counted for each sample. n.s. indicates not significant, ****$P < 0.0001$, one-way ANOVA test.

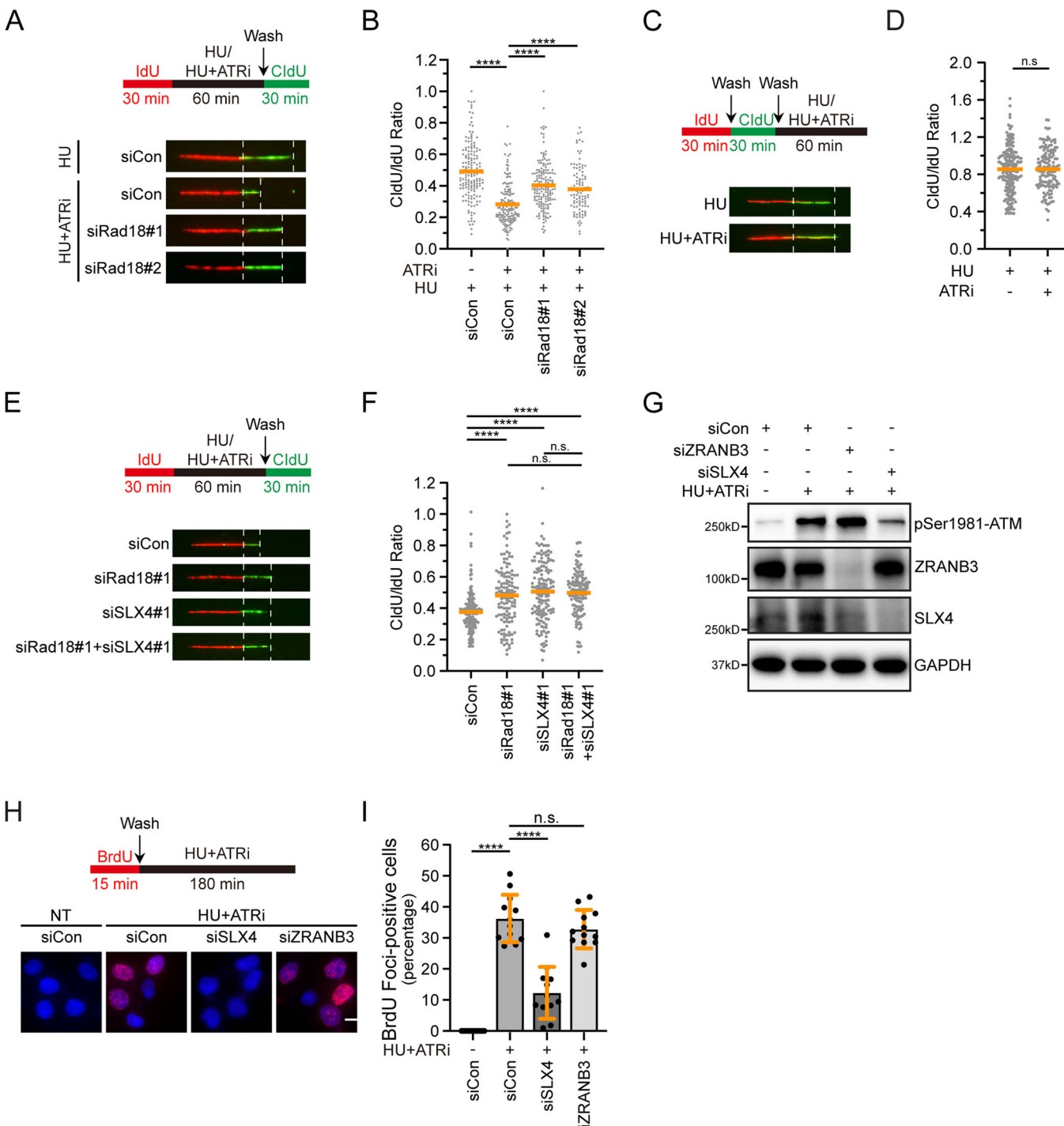

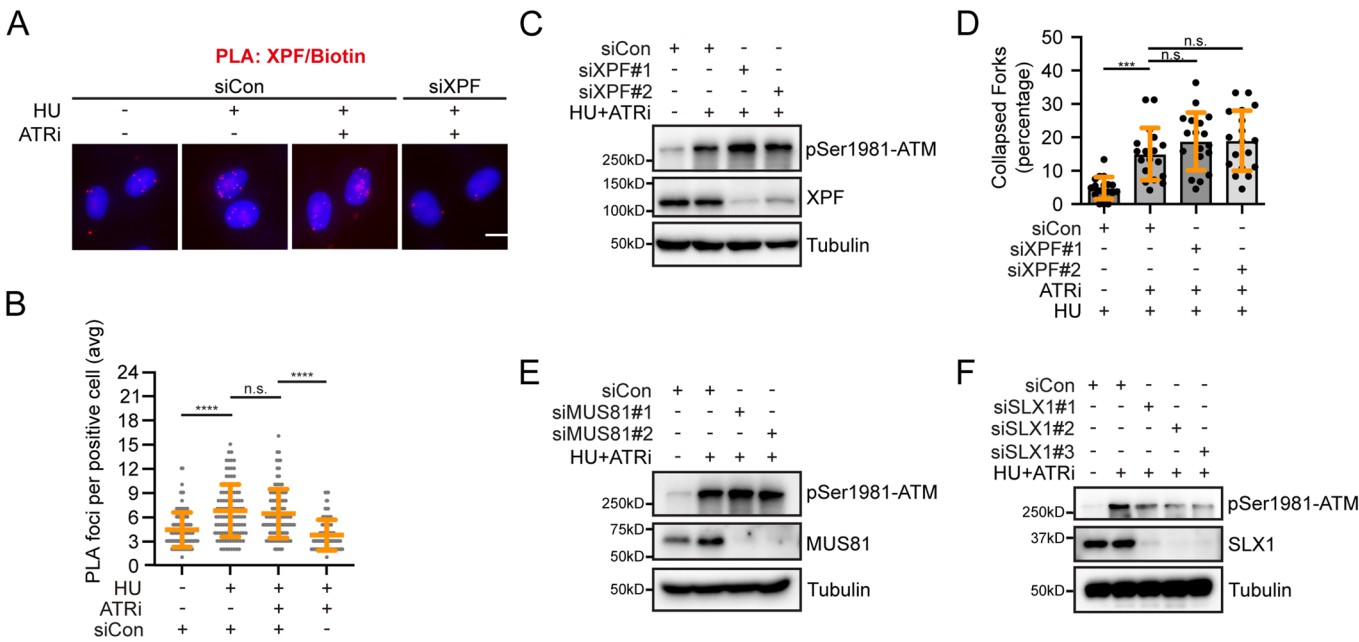

**Figure EV2.  SLX1 is responsible for SLX4-dependent stalled fork collapse following ATR inhibition.**

(A, B) XPF accumulates at the stalled fork. HeLa cells were pulse-labeled with 10 μM EdU for 15 min, and then treated with 2 mM HU alone or in combination with 2 μM VE-821 for 1 h. Cells were then subjected to PLA with anti-XPF and anti-biotin antibodies. Representative images of PLA foci (red) (A). DNA was stained with DAPI. Scale bar, 10 μm. Quantification of PLA foci number per focus-positive cell (B). Data represent means ± SD of three independent experiments. More than 100 cells were counted for each sample. n.s., not significant, ****$P < 0.0001$, one-way ANOVA test. (C) HeLa cells were transfected with the siRNAs for XPF. After 48 h of transfection, cells were treated with 2 mM HU and 2 μM VE-821 for 3 h. Cell lysates were then prepared and western blot analysis was carried out as indicated. (D) HeLa cells were transfected with the siRNAs for XPF. After 48 h of transfection, cells were incubated with 50 μM IdU for 30 min, treated with 2 mM HU and 2 μM VE-821 for 1 h, and incubated with 100 μM CldU for 30 min. Each black dot in the graph represents the percentage of collapsed forks in each measurement, and more than 200 fibers were measured for each sample. Data represent means ± SD of three independent experiments. n.s., not significant, ***$P < 0.001$, one-way ANOVA test. (E) MUS81 is not responsible for the fork collapse in the absence of ATR under replication stress. HeLa cells were transfected with the siRNAs for MUS81. After 48 h of transfection, cells were treated with 2 mM HU and 2 μM VE-821 for 3 h. Cell lysates were then prepared and western blot analysis was carried out as indicated. (F) SLX1 plays a role in fork collapse in the absence of ATR under replication stress. HeLa cells were transfected with the siRNAs for SLX1. After 48 h of transfection, cells were treated with 2 mM HU and 2 μM VE-821 for 3 h. Cell lysates were then prepared and western blot analysis was carried out as indicated.

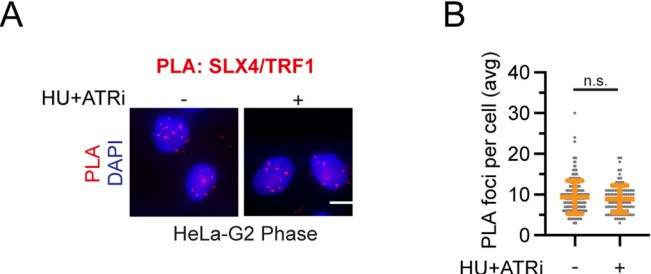

**A**

**PLA: SLX4/TRF1**

HeLa-G2 Phase

**B**

HU+ATRi

**Figure EV3. ATR inhibition does not affect SLX4 accumulation at telomeres in ALT-negative HeLa cells in the G2 phase.**

(A, B) For Hela cells enriched in the G2 phase, cells were treated with thymidine for 20 h, released into DMEM for 4 h, and then treated with CDK1 inhibitor, 10 µM RO-3306, for 15 h. The synchronized cells were mock-treated or treated with 2 mM HU and 2 µM VE-821 for 3 h, subjected to PLA using anti-SLX4 and anti-TRF1 antibodies. Representative images of PLA foci (A). Scale bar, 10 µm. Quantification of PLA foci number per focus-positive cell (B). Data represent means ± SD of three independent experiments. More than 100 cells were counted for each sample. n.s. indicates not significant, one-way ANOVA test.

