## [Peer Review File · The EMBO Journal]

ATR limits Rad18-mediated PCNA monoubiquitination to preserve replication fork stability

Jun Huang, Siyuan Chen, Ting Liu, and Chen Pan

Corresponding author(s): Jun Huang (jhuang@zju.edu.cn) , Ting Liu (liuting518@zju.edu.cn)

Review Timeline:

Submission Date:	14th Sep 23
Editorial Decision:	4th Oct 23
Revision Received:	19th Jan 24
Editorial Decision:	9th Feb 24
Revision Received:	15th Feb 24
Accepted:	26th Feb 24

Editor: Hartmut Vodermaier

Transaction Report:

Prof. Jun Huang
Zhejiang University
Life Sciences Institute
388 Yuhangtang Road
Hangzhou, Zhejiang 310058
China

4th Oct 2023

Re: EMBOJ-2023-115612
ATR limits Rad18-mediated PCNA monoubiquitination to preserve replication fork stability

Dear Dr. Huang,

Thank you for submitting your study on ATR-Rad18 crosstalk to The EMBO Journal. It has now been assessed by three expert referees, whose reports are copied below for your information. As you will see, all referees appreciate the interest and potential importance of your findings. At the same time, they are however not yet convinced that all main conclusions are adequately supported by the current data, and that alternative scenarios have been decisively addressed/ruled out.

Should you be able to satisfactorily address the key issues raised in all three reports, we would be happy to consider a revised manuscript further for publication. Since it is our policy to consider only a single round of major revision, it is important to fully answer to all comments at the time of resubmission - I would therefore invite you to get back to me with a tentative response letter/revision plan already during the early stages of the revision work. On the basis of this response, we could then further discuss the revision requirements and how to best address the key concerns. I should add that we could also offer extension of the default three-months revision period if needed, with our 'scooping protection' (meaning that competing work appearing elsewhere in the meantime will not affect our considerations of your study) remaining of course valid also throughout this extension.

Detailed information on preparing, formatting and uploading a revised manuscript can be found below and in our Guide to Authors. Thank you again for the opportunity to consider this work for The EMBO Journal, and I look forward to hearing from you in due time.

Yours sincerely,

Hartmut Vodermaier

9) Digital image enhancement is acceptable practice, as long as it accurately represents the original data and conforms to community standards. If a figure has been subjected to significant electronic manipulation, this must be clearly noted in the figure legend and/or the 'Materials and Methods' section. The editors reserve the right to request original versions of figures and the original images that were used to assemble the figure. Finally, we generally encourage uploading of numerical as well as gel/blot image source data; for details see: embopress.org/page/journal/14602075/authorguide#sourcedata

At EMBO Press, we ask authors to provide source data for the main manuscript figures. Our source data coordinator will contact you to discuss which figure panels we would need source data for and will also provide you with helpful tips on how to upload and organize the files.

In the interest of ensuring the conceptual advance provided by the work, we recommend submitting a revision within 3 months (2nd Jan 2024). Please discuss the revision progress ahead of this time with the editor if you require more time to complete the revisions. Use the link below to submit your revision:

Link Not Available

Referee #1:

In this manuscript, the authors verify and characterize an ATR-dependent phosphorylation site in the human ubiquitin ligase RAD18. The site had been previously identified and in fact, an antibody against this site is available. The authors unexpectedly show that this site is associated with a PCNA-interacting motif in RAD18, and phosphorylation abolishes interaction with PCNA. As a consequence, activity of RAD18 towards PCNA is limited. The authors postulate that this regulatory device limits excessive accumulation of SLX4, a nuclease that would otherwise destabilize stalled forks. They claim that SLX4 is a specific interactor of monoubiquitylated PCNA. Finally, they postulate a similar mechanism to operate at ALT-mediated telomere lengthening.

The findings about a phosphorylation-regulated PCNA interaction motif in RAD18 are interesting and significant, even though this phenomenon seems to be conserved only among primates. Although the phosphorylation site itself has been known before, its regulatory potential described in this manuscript is novel and could potentially form the basis of a new layer of regulation of the DNA damage bypass pathway.

At the same time, some of the authors' major claims are not supported by experimental data and could be explained by alternative models. Major concerns are as follows:

1. Consistent with RAD18's activity towards PCNA, the authors postulate a regulatory mechanism that specifically impinges on PCNA monoubiquitylation. However, they do not consider the possibility that the observed consequences are due to an indirect effect resulting from further modification of PCNA, i.e., specifically, PCNA polyubiquitylation. PCNA polyubiquitylation has now

been implicated in the ALT pathway in two manuscripts available on bioRxiv (10.1101/2023.07.13.548953v1). At the replication fork, PCNA polyubiquitylation is thought to drive fork reversal, and SLX4 is known to be involved in the degradation of reversed forks. It would therefore make sense to postulate that excessive RAD18 activity would not act directly but would rather cause excessive polyubiquitylation of PCNA, followed by excessive fork reversal and therefore excessive vulnerability to SLX4. This would not require any direct interaction between PCNA-Ub and SLX4. The authors would need to do experiments to exclude (or confirm) this alternative scenario. For example, they could repeat key experiments under conditions that prevent fork reversal (SMARCAL1, HLTf, ZRANB3 knockdown) or PCNA polyubiquitylation.

2. The data do not show that SLX4 is recruited by PCNA monoubiquitylation. As explained above, its activity could be an indirect consequence of fork reversal. Moreover, the *in vitro* data are insufficient. They pretty much only show that the UBZ4 domain of SLX4 indeed binds to ubiquitin. Their assays do not dissect whether SLX4 has a basal affinity for PCNA and whether it specifically recognizes the monoubiquitylated form. Thus, it is unclear whether monoubiquitylated PCNA is really a physiological interaction partner of SLX4.

3. In Figure 2D, it would be very helpful to indicate not only the percentage of collapsed forks, but also IdU and CldU tract lengths. Recently, ATR has been shown to protect replication forks from nucleolytic degradation upon HU treatment (Leung 2023, Cell Rep). Do the authors observe this in their setup? Or is 60 min a too short time point to observe this phenotype? Here, comparison of CldU tract lengths would give important information regarding those forks that have restarted - do they move with the same velocity as before the treatment or is the replication speed altered after the restart?

4. Considering that the RAD18 phosphorylation site is not conserved outside of primates, it is unclear how significant this regulatory mechanism is. Is there any equivalent or analogous scenario in other organisms? The authors could perform some phenotypic assays with ATR in mouse cells and demonstrate that ATR has no effect here, given that RAD18 lacks the phosphorylation site.

Minor Comments

1. Figure 2A - why is GAPDH (a soluble protein) used as a loading control for the chromatin fraction?

2. Figure 7 - the lower part of the model is confusing.

3. In their introduction and also sometimes in the results part, the authors mix previous findings from yeast and human cells. Given that their observed phenomenon is not conserved at all, they should be a lot more careful when basing their experiments on prior findings in other organisms. At least they should always explicitly state what organism is concerned when providing information about RAD18.

Referee #2:

In this manuscript, the authors propose a novel molecular mechanism by which ATR stabilizes replication forks under HU-induced stress. Firstly, ATR phosphorylates Rad18. The phosphorylation impairs interaction between Rad18 and PCNA, leading to reduction in PCNA monoubiquitination. They further suggest that monoubiquitinated PCNA may serve as a platform for recruiting SLX4, a crucial scaffold protein acting with three endonucleases: XPF, SLX1, and Mus81. Therefore, in the suggested model, if ATR function is compromised, monoubiquitinated PCNA levels are inappropriately upregulated, leading to concomitant increase in recruitment of SLX4-based nucleases. Consequently, SLX4-based nucleases may excessively attack the forks, resulting in fork collapse.

This is a novel story and holds potential interest for the field. Overall, the manuscript is well written. However, although most of the data appear to support the authors' conclusion, there are some unclear data (see below). Furthermore, additional experiments should be required to strengthen the conclusion.

Major points

1. I think one important issue is not addressed at all; namely possible involvement of XPF endonuclease, a partner of SLX4, in the process the authors investigated. As mentioned in "Discussion", it is reported that SLX4-XPF is recruited to ICL sites via K63-linked polyubiquitin chains generated by RNF168 (Katsuki et al., Cell Reports 2021). In addition, Bétouset et al. report that the fork breakage is induced by XPF (and Artemis) in cells undergoing HU-induced replication stress (Plos Genetics e1007541, 2018). First, this paper should be cited and discussed in detail in the manuscript. In their experimental setting, silencing of XPF increases stalled forks without restart (Bétouset et al., Plos Genetics 2018). On the other hand, in the current manuscript, it is shown that SLX4 silencing rather decreases collapsed forks (Figure 2E). It is naturally possible that SLX4 silencing does not necessarily phenocopy XPF silencing in HU-treated cells. Nevertheless, the issue should be investigated in the experimental system the authors adopt. Firstly, the dynamics of XPF recruitment to the stalled forks, for example by PLA assay used for SLX4 (Figure 4A and B), should be investigated and should be compared with that of SLX4. Secondly, the effect of XPF silencing on pSer1981-ATM (as a marker of fork breakage) and fork restart should be examined and should be compared with the effect of SLX4 silencing (Figure 2E and Figure 4H).

2. In this context, it would be also better to investigate the effect of SLX1 and Mus81 silencing on pSer1981-ATM with comparing

with SLX4 (Figure 4H).

3. Figure 3D-F: The interaction between Rad18 and PCNA and the roles of S403 and the PIP box in this interaction is shown only by pull down assays with purified protein. If co-immunoprecipitation assays will also provide similar findings, they support the authors' conclusion more rigorously.
4. Figure 5B: In the suggested model, SLX4 interacts with ubiquitinated PCNA. However, in Figure 5B, a large amount of non-ubiquitinated PCNA co-precipitates with SFB-SLX4. Why? In addition, control IP experiments with control cells without SFB-SLX4 should be presented.
5. Figure 5D: In relation to the point mentioned above, it should be clarified whether non-ubiquitinated GST-PCNA is pulled down with MBP-UBZ4 or not.
6. The authors propose that the ATR-Rad18-PCNA-SLX4 axis may also function in telomere stability in ALT cells (Figure 6). In this regard, Zhang et al. just recently report that Rad18-dependent PCNA ubiquitination may recruit SNM1A nuclease to maintain ALT (Nature 619, 201, 2023). This paper should be cited and discussed in the manuscript. According to the data presented in this paper, when Rad18 is inhibited, recruitment of SNM1A to telomeres is compromised, while SLX4 recruitment remains unaffected. This point especially should be discussed in detail.

Minor points

1. Figure 2A: It would be better to also show the kinetics of ATM phosphorylation and PCNA ubiquitination in HeLa cells treated only with HU.
2. Figure 2E and H: It is unclear how these graphs are depicted. What do the black dots mean?
3. Figure 3D and E: Control experiments to show the background precipitation of PCNA (and RPA in Figure 3D) in the absence of His-sumo-Rad18 should be presented.
4. Figure 3F: The labeling "GST Pulldown" is wrong. It would be "MBP Pulldown".
5. Throughout the manuscript, data should be presented with mean {plus minus} SD (not {plus minus} SEM).
6. Statistical analysis: It is stated that one-way ANOVA test is used for the comparison of more than two groups. However, it is unclear what kind of post-hoc test is used for multiple comparisons.
7. PLA assays: It should be clarified how many cells were counted in each experiment.
8. Methods: Some descriptions for the method by which siRNA and plasmid is co-transfected into cells should be presented.

Referee #3:

The manuscript "ATR limits Rad18-mediated PCNA monoubiquitination to preserve replication fork and telomerase-independent telomere stability" by Chen et al., explored the role of ATR-dependent regulation of PCNA ubiquitylation that mediates replication stress-induced genome instability. The authors show that ATR-dependent phosphorylation prevents PCNA ubiquitylation during replication stress, and DSBs arising after HU+ATRi treatments are mediated by SLX4 recruited to ubiquitylated PCNA. They also demonstrate that this mechanism acts specifically at ALT telomeres where ATR activity is also essential to prevent excessive SLX4 accumulation. These are very important findings for our understanding of the replication stress response, as well as for the informed clinical use of ATR inhibitors, especially for the treatment of ALT tumors. The article should be published after the points listed below are addressed:

- My main concern is about the effects observed in G2 phase on ALT telomeres. Since the majority of DNA replication is completed at the time (in agreement with BrdU FACS on 6a), and only some telomeres continue some repair/replication processes, I would expect that the amount of PCNA on chromatin should be very low. Yet, on 6C we see equal amounts of PCNA on chromatin in S and G2 cells, and similar ubiquitylation levels in response to ATRi+HU. Additionally, HU acts by inhibiting RNR in S-phase when nucleotide consumption rate is extremely high, so the cells run out of them very quickly resulting in stalled forks. In G2, there's barely any DNA synthesis, so I would not expect much of an effect of HU on DNA synthesis under the same conditions. In my opinion, additional explanations are required for these results. In order to confirm that these effects are indeed ALT-specific, a non-ALT cell line (HeLa, for example) should be used, and the authors need to show that these effects in G2 are absent in non-ALT cells. Also, showing TRF1/SLX4 PLA for S-phase and G1 phase cells could help confirm that the effects are ALT-associated.
- On panel 1F adding CDC7 inhibitor to block origin firing seems to have increased PCNA ubiquitylation. CDC7i should

decrease the number of active replication forks and the amount of PCNA on chromatin, so this is a surprising observation and should be discussed.

- On page 5 authors state "Upon activation, ATR activates its downstream kinases, including Chk1 and Wee1, to inhibit cell-cycle progression, suppress late origin firing, and stabilize stalled forks (Saldivar et al., 2017)." To my knowledge, there is no data indicating that ATR activates WEE1 in mammalian cells, and the cited review does not claim that either.

We express our gratitude to all the reviewers for their careful and constructive comments on our manuscript. Over the last three months, we have conducted a series of experiments as suggested by the reviewers. We believe that these new experimental findings effectively address the reviewers' concerns and offer additional support to our main conclusions. Below is our point-by-point response to the reviewers' critiques:

Referee #1:

In this manuscript, the authors verify and characterize an ATR-dependent phosphorylation site in the human ubiquitin ligase RAD18. The site had been previously identified and in fact, an antibody against this site is available. The authors unexpectedly show that this site is associated with a PCNA-interacting motif in RAD18, and phosphorylation abolishes interaction with PCNA. As a consequence, activity of RAD18 towards PCNA is limited. The authors postulate that this regulatory device limits excessive accumulation of SLX4, a nuclease that would otherwise destabilize stalled forks. They claim that SLX4 is a specific interactor of monoubiquitylated PCNA. Finally, they postulate a similar mechanism to operate at ALT-mediated telomere lengthening.

The findings about a phosphorylation-regulated PCNA interaction motif in RAD18 are interesting and significant, even though this phenomenon seems to be conserved only among primates. Although the phosphorylation site itself has been known before, its regulatory potential described in this manuscript is novel and could potentially form the basis of a new layer of regulation of the DNA damage bypass pathway.

Thank you for your constructive suggestions and comments.

At the same time, some of the authors' major claims are not supported by experimental data and could be explained by alternative models. Major concerns are as follows:

1. Consistent with RAD18's activity towards PCNA, the authors postulate a regulatory mechanism that specifically impinges on PCNA monoubiquitylation. However, they do not consider the possibility that the observed consequences are due to an indirect effect resulting from further modification of PCNA, i.e., specifically, PCNA polyubiquitylation. PCNA polyubiquitylation has now been implicated in the ALT pathway in two manuscripts available on bioRxiv (10.1101/2023.07.13.548953v1). At the replication fork, PCNA polyubiquitylation is thought to drive fork reversal, and SLX4 is known to be involved in the degradation of reversed forks. It would therefore make sense to postulate that excessive RAD18 activity would not act directly but would rather cause excessive polyubiquitylation of PCNA, followed by excessive fork reversal and therefore excessive vulnerability to SLX4. This would not require any direct interaction between PCNA-Ub and SLX4. The authors would need to do experiments to exclude (or confirm) this alternative scenario. For example, they could repeat key experiments under conditions that prevent fork reversal (SMARCAL1, HLTF, ZRANB3 knockdown) or PCNA

polyubiquitylation.

We appreciate the insightful comments and suggestions provided by the reviewer. As highlighted by the reviewer, ZRANB3, a DNA translocase, is known to interact with polyubiquitinated PCNA, facilitating replication fork reversal. To examine the significance of PCNA polyubiquitination-dependent ZRANB3-mediated fork reversal in the context of SLX4-dependent fork collapse following HU and ATRi treatment, we conducted experiments depleting ZRANB3 and assessed its impact on ATM phosphorylation and BrdU foci formation. As shown in Supplementary Figure 1G-1I, ZRANB3 knockdown did not significantly affect ATM phosphorylation and BrdU foci formation in cells treated with HU and ATRi. These results suggest that PCNA polyubiquitination-dependent ZRANB3-mediated fork reversal may not be a prerequisite for the observed fork collapse under these conditions.

2. The data do not show that SLX4 is recruited by PCNA monoubiquitylation. As explained above, its activity could be an indirect consequence of fork reversal. Moreover, the in vitro data are insufficient. They pretty much only show that the UBZ4 domain of SLX4 indeed binds to ubiquitin. Their assays do not dissect whether SLX4 has a basal affinity for PCNA and whether it specifically recognizes the monoubiquitylated form. Thus, it is unclear whether monoubiquitylated PCNA is really a physiological interaction partner of SLX4.

We appreciate the insightful comment and suggestions provided by the reviewer. Considering that a substantial amount of non-ubiquitinated PCNA co-precipitates with SLX4 regardless of replication stress, we speculated that SLX4 may contain potential PIP boxes mediating its basal affinity for PCNA. Upon scrutinizing the amino acid sequence of the SLX4 protein, we identified three potential PIP boxes in the N-terminal region of SLX4 (Figure 5A). Strikingly, deletion of the first (amino acid 58-59) or second (amino acid 204-216) putative PIP box, but not the third (amino acid 267-277), significantly disrupted the interaction between SLX4 and PCNA (Figure 5C). These results suggest that both the UBZ4-1 domain and the PIP boxes may specifically mediate the interaction between SLX4 and the ubiquitinated PCNA under conditions of replication stress. The results of these experiments were integrated into the revised manuscript to provide a more comprehensive understanding of the underlying mechanisms. We once again express our gratitude for the valuable input from the reviewer, which has significantly enhanced the quality and depth of our study.

3. In Figure 2D, it would be very helpful to indicate not only the percentage of collapsed forks, but also IdU and CldU tract lengths. Recently, ATR has been shown to protect replication forks from nucleolytic degradation upon HU treatment (Leung 2023, Cell Rep). Do the authors observe this in their setup? Or is 60 min a too short time point to observe this phenotype? Here,

comparison of CldU tract lengths would give important information regarding those forks that have restarted - do they move with the same velocity as before the treatment or is the replication speed altered after the restart?

Thank you for your suggestion. A recent insightful study by Leung (2023, Cell Rep) has demonstrated that ATR protects replication forks from nucleolytic degradation upon HU treatment. In our experiments, we did not observe this phenomenon (Supplementary Figure 1C-1D), and we concur with the reviewer's assessment that the 60-minute time point may be too short to capture this particular phenotype. Additionally, as you suggested, we also performed DNA fiber assays to evaluate the efficiency of restart by measuring the ratio of CldU to IdU track lengths. As shown in Supplementary Figure 1A-1B, treatment with ATR inhibitor resulted in a substantial decrease in the ratio of CldU to IdU track lengths, indicating that ATR inactivation hampered the restart of stalled replication forks. Strikingly, depletion of Rad18 largely reversed the defects in fork restart caused by ATR inhibition (Supplementary Figure 1A-1B). More importantly, simultaneous depletion of Rad18 and SLX4 did not cause a further increase in the ratio of CldU to IdU track lengths (Supplementary Figure 1E-1F), indicating a functional interdependency between Rad18 and SLX4 within the same pathway.

4. Considering that the RAD18 phosphorylation site is not conserved outside of primates, it is unclear how significant this regulatory mechanism is. Is there any equivalent or analogous scenario in other organisms? The authors could perform some phenotypic assays with ATR in mouse cells and demonstrate that ATR has no effect here, given that RAD18 lacks the phosphorylation site.

Thank you for your suggestion. We have also observed an analogous scenario in other proteins, such as ATR. The group led by Dr. David Cortez reported that Thr-1989 phosphorylation serves as a marker of active ATR, and this phosphorylation site is conserved only in primates (Nam et al, 2011).

In response to your recommendation, we conducted phenotypic assays with ATRi in mouse cells to explore whether ATRi has an effect in the absence of the phosphorylation site in RAD18. As shown in Figure 3I-3J, inhibiting ATR did not further enhance replication stress-induced PCNA monoubiquitination in both MEF and NIH3T3 cell lines. As per your recommendation, we also investigated whether ATR inhibition affects replication stress-induced fork collapse in these two murine cell lines. As shown in Figure 3K-3L, inhibiting ATR did not further enhance replication stress-induced fork collapse in both cell lines (as our pSer1981-ATR antibody only recognizes human ATR, here we used the levels of γ -H2AX as an indicator of fork collapse). The results of these experiments were integrated into the revised manuscript. This observation strengthens the notion of higher primate-specific ATR-mediated regulation of PCNA monoubiquitination through Ser403

phosphorylation in human Rad18.

Minor Comments

1. Figure 2A - why is GAPDH (a soluble protein) used as a loading control for the chromatin fraction?

We apologize for the confusion. In the original Figure 2A, we used GAPDH as a loading control for the detection of pSer1981-ATM in whole cell lysate (the ATM phosphorylation pattern was consistent in the whole cell lysate, soluble fraction, and chromatin fraction). For the detection of Ub-PCNA in the chromatin fraction, non-ubiquitinated PCNA was used as the loading control. We have now repeated these experiments and used the H3 as a loading control for chromatin fraction (Please see the revised Figure 2A).

2. Figure 7 - the lower part of the model is confusing.

We apologize for any confusion caused by the model presented in Figure 7. We carefully revised the lower part of the model in the revised manuscript to ensure it is more understandable.

3. In their introduction and also sometimes in the results part, the authors mix previous findings from yeast and human cells. Given that their observed phenomenon is not conserved at all, they should be a lot more careful when basing their experiments on prior findings in other organisms. At least they should always explicitly state what organism is concerned when providing information about RAD18.

We apologize for any confusion and appreciate the reviewer's attention to this matter. In the revised manuscript, we have emphasized that ATR phosphorylates human RAD18 in the abstract, introduction, and results sections.

Referee #2:

In this manuscript, the authors propose a novel molecular mechanism by which ATR stabilizes replication forks under HU-induced stress. Firstly, ATR phosphorylates Rad18. The phosphorylation impairs interaction between Rad18 and PCNA, leading to reduction in PCNA monoubiquitination. They further suggest that monoubiquitinated PCNA may serve as a platform for recruiting SLX4, a crucial scaffold protein acting with three endonucleases: XPF, SLX1, and Mus81. Therefore, in the suggested model, if ATR function is compromised, monoubiquitinated PCNA levels are inappropriately upregulated, leading to concomitant increase in recruitment of SLX4-based nucleases. Consequently, SLX4-based nucleases may excessively attack the forks, resulting in fork collapse.

This is a novel story and holds potential interest for the field. Overall, the manuscript is well written. However, although most of the data appear to support the authors' conclusion, there are some unclear data (see below). Furthermore, additional experiments should be required to strengthen the conclusion.

Thank you for your constructive suggestions and comments.

Major points

1. I think one important issue is not addressed at all; namely possible involvement of XPF endonuclease, a partner of SLX4, in the process the authors investigated. As mentioned in "Discussion", it is reported that SLX4-XPF is recruited to ICL sites via K63-linked polyubiquitin chains generated by RNF168 (Katsuki et al., Cell Reports 2021). In addition, Bétouset et al. report that the fork breakage is induced by XPF (and Artemis) in cells undergoing HU-induced replication stress (Plos Genetics e1007541, 2018). First, this paper should be cited and discussed in detail in the manuscript. In their experimental setting, silencing of XPF increases stalled forks without restart (Bétouset et al., Plos Genetics 2018). On the other hand, in the current manuscript, it is shown that SLX4 silencing rather decreases collapsed forks (Figure 2E). It is naturally possible that SLX4 silencing does not necessarily phenocopy XPF silencing in HU-treated cells. Nevertheless, the issue should be investigated in the experimental system the authors adopt. Firstly, the dynamics of XPF recruitment to the stalled forks, for example by PLA assay used for SLX4 (Figure 4A and B), should be investigated and should be compared with that of SLX4. Secondly, the effect of XPF silencing on pSer1981-ATM (as a marker of fork breakage) and fork restart should be examined and should be compared with the effect of SLX4 silencing (Figure 2E and Figure 4H).

Thank you for your valuable suggestion. As recommended, we included and described these references in the revised manuscript. Additionally, we performed PLA experiments to investigate the dynamics of XPF recruitment to stalled forks. As shown in Supplementary Figure 2A-2B, HU stimulation led to the accumulation of XPF at stalled forks; however, ATR inhibition did not further increase this enrichment. Additionally, in contrast to SLX4

depletion, the absence of XPF did not suppress ATM phosphorylation or the frequency of fork collapse following HU and ATRi treatment (**Supplementary Figure 2C-2D**).

2. In this context, it would be also better to investigate the effect of SLX1 and Mus81 silencing on pSer1981-ATM with comparing with SLX4 (Figure 4H).

Thank you for your valuable suggestion. As shown in **Supplementary Figure 2E, despite MUS81 being responsible for the formation of DSBs after prolonged exposure to replication inhibitors, we found that depleting MUS81 had no effect on ATM phosphorylation following HU and ATRi treatment. However, depletion of SLX1 significantly suppressed ATM phosphorylation under the same conditions, suggesting that the SLX1 endonuclease may play a key role in SLX4-dependent stalled fork collapse following ATR inhibition (**Supplementary Figure 2F**). We once again express our gratitude for the valuable input from the reviewer, which has significantly enhanced the quality and depth of our study.**

3. Figure 3D-F: The interaction between Rad18 and PCNA and the roles of S403 and the PIP box in this interaction is shown only by pull down assays with purified protein. If co-immunoprecipitation assays will also provide similar findings, they support the authors' conclusion more rigorously.

Thank you for your valuable suggestion. We conducted Co-IP experiments, as recommended by the reviewer, to further substantiate the interaction between Rad18 and PCNA and the roles of S403 and the PIP box in this interaction. As shown in **Figure 3F, compared with wild-type RAD18, the S403A mutant, S403E mutant, or PIP motif deletion mutant exhibited reduced efficiency in binding to PCNA.**

4. Figure 5B: In the suggested model, SLX4 interacts with ubiquitinated PCNA. However, in Figure 5B, a large amount of non-ubiquitinated PCNA co-precipitates with SFB-SLX4. Why? In addition, control IP experiments with control cells without SFB-SLX4 should be presented.

We appreciate the insightful comment and suggestions provided by the reviewer. Considering that a substantial amount of non-ubiquitinated PCNA co-precipitates with SLX4 regardless of replication stress, we speculated that SLX4 may contain potential PIP boxes mediating its basal affinity for PCNA. Upon scrutinizing the amino acid sequence of the SLX4 protein, we identified three potential PIP boxes in the N-terminal region of SLX4 (Figure 5A**). Strikingly, deletion of the first (amino acid 58-59) or second (amino acid 204-216) putative PIP box, but not the third (amino acid 267-277), significantly disrupted the interaction between SLX4 and PCNA (**Figure 5C**). These results suggest that both the UBZ4-1 domain and the PIP boxes may specifically mediate the interaction between SLX4 and the ubiquitinated PCNA under conditions of replication stress. The results of these experiments were integrated into the revised manuscript to provide a more comprehensive**

understanding of the underlying mechanisms. We once again express our gratitude for the valuable input from the reviewer, which has significantly enhanced the quality and depth of our study.

Additionally, as suggested by the reviewer, we included control IP experiments using cells that do not express SFB-SLX4 (please see the revised Figure 5B). These results were integrated into the revised manuscript.

5. Figure 5D: In relation to the point mentioned above, it should be clarified whether non-ubiquitinated GST-PCNA is pulled down with MBP-UBZ4 or not.

Thank you for your suggestion. We conducted pulldown experiments and observed that non-ubiquitinated PCNA cannot be pulled down with MBP-UBZ4. We have included these results in the revised manuscript (please see the revised Figure 5D-5E).

6. The authors propose that the ATR-Rad18-PCNA-SLX4 axis may also function in telomere stability in ALT cells (Figure 6). In this regard, Zhang et al. just recently report that Rad18-dependent PCNA ubiquitination may recruit SNM1A nuclease to maintain ALT (Nature 619, 201, 2023). This paper should be cited and discussed in the manuscript. According to the data presented in this paper, when Rad18 is inhibited, recruitment of SNM1A to telomeres is compromised, while SLX4 recruitment remains unaffected. This point especially should be discussed in detail.

Thank you for your valuable suggestion. We have now included a citation for the paper by Zhang et al. and discussed it in the revised manuscript.

Minor points

1. Figure 2A: It would be better to also show the kinetics of ATM phosphorylation and PCNA ubiquitination in HeLa cells treated only with HU.

Thank you for your suggestion. We performed the experiment as suggested by the reviewer to show the kinetics of ATM phosphorylation and PCNA ubiquitination in HeLa cells treated only with HU. This data was included in the revised manuscript (please see the revised Figure 2A).

2. Figure 2E and H: It is unclear how these graphs are depicted. What do the black dots mean?

We apologize for any confusion. In Figure 2E, each black dot in the graph represents the percentage of stalled forks in each measurement, and more than 200 fibers were measured for each sample. In Figure 2H, similarly, each black dot in the graph represents the percentage of BrdU-positive cells in each measurement, with more than 300 cells were

counted in each experiment. These experiments were repeated at least three times, yielding similar results. This information was clarified in the revised Figure legends.

3. Figure 3D and E: Control experiments to show the background precipitation of PCNA (and RPA in Figure 3D) in the absence of His-sumo-Rad18 should be presented.

Thank you for your suggestion. We repeated the pulldown experiments, including control experiments to show the background precipitation of PCNA and RPA in the absence of His-sumo-Rad18 (please see the revised Figure 3D-3F).

4. Figure 3F: The labeling "GST Pulldown" is wrong. It would be "MBP Pulldown".

We apologize for the mislabeling in Figure 3F and appreciate your attention to detail. The error was corrected in the revised Figure (please see the revised Figure 3G).

5. Throughout the manuscript, data should be presented with mean {plus minus} SD (not {plus minus} SEM).

Thank you for your suggestion. We presented the data with mean \pm SD instead of mean \pm SEM in the revised Figures.

6. Statistical analysis: It is stated that one-way ANOVA test is used for the comparison of more than two groups. However, it is unclear what kind of post-hoc test is used for multiple comparisons.

We apologize for any lack of clarity. For the statistical analysis and comparison of the groups, we used one-way ANOVA with Dunnett's post hoc test, performed using GraphPad Prism Version 9. This information was now included in the Quantification and Statistical Analysis section.

7. PLA assays: It should be clarified how many cells were counted in each experiment.

We apologize for any lack of clarity. We counted more than 100 PLA foci-positive cells for each sample in the PLA assays.

8. Methods: Some descriptions for the method by which siRNA and plasmid is co-transfected into cells should be presented.

We apologize for any lack of clarity. Cells were transfected with the indicated siRNAs using Lipofectamine RNAiMAX transfection reagent (Invitrogen) following the manufacturer's instructions. 48 hours after siRNA transfection, cells were repassaged and subsequently

transfected with the indicated plasmids using Hieff Trans™ Liposomal Transfection Reagent following the manufacturer's guidelines (Yeasen Biotechnology). We have now included this information in the revised manuscript for improved clarity.

Referee #3:

The manuscript "ATR limits Rad18-mediated PCNA monoubiquitination to preserve replication fork and telomerase-independent telomere stability" by Chen et al., explored the role of ATR-dependent regulation of PCNA ubiquitylation that mediates replication stress-induced genome instability. The authors show that ATR-dependent phosphorylation prevents PCNA ubiquitylation during replication stress, and DSBs arising after HU+ATRi treatments are mediated by SLX4 recruited to ubiquitylated PCNA. They also demonstrate that this mechanism acts specifically at ALT telomeres where ATR activity is also essential to prevent excessive SLX4 accumulation. These are very important findings for our understanding of the replication stress response, as well as for the informed clinical use of ATR inhibitors, especially for the treatment of ALT tumors. The article should be published after the points listed below are addressed:

Thank you for your constructive suggestions and comments.

- My main concern is about the effects observed in G2 phase on ALT telomeres. Since the majority of DNA replication is completed at the time (in agreement with BrdU FACS on 6a), and only some telomeres continue some repair/replication processes, I would expect that the amount of PCNA on chromatin should be very low. Yet, on 6C we see equal amounts of PCNA on chromatin in S and G2 cells, and similar ubiquitylation levels in response to ATRi+HU. Additionally, HU acts by inhibiting RNR in S-phase when nucleotide consumption rate is extremely high, so the cells run out of them very quickly resulting in stalled forks. In G2, there's barely any DNA synthesis, so I would not expect much of an effect of HU on DNA synthesis under the same conditions. In my opinion, additional explanations are required for these results. In order to confirm that these effects are indeed ALT-specific, a non-ALT cell line (HeLa, for example) should be used, and the authors need to show that these effects in G2 are absent in non-ALT cells. Also, showing TRF1/SLX4 PLA for S-phase and G1 phase cells could help confirm that the effects are ALT-associated.

Thank you for bringing up this concern. The reviewer is correct in pointing out that there's a substantial difference in PCNA levels on chromatin between G2 and S phases. In original Figure 6C, we made adjustments to cell lysates to ensure that PCNA levels were comparable, aiming to facilitate a direct comparison of ATR inhibition's impact on PCNA ubiquitination between these phases. In the revised manuscript, we repeated the experiment without making adjustments to cell lysates, allowing for a more accurate representation of the distinctions between these phases (please see the revised Figure 6C).

Additionally, as per the reviewer's suggestion, we performed PLA experiments to visualize the colocalization of SLX4 with TRF1 in G1, S, and G2 phases. As shown in Figure 6D-6E, co-treatment with HU and ATRi substantially increased the number of SLX4/TRF1 PLA

foci specifically in G2 phase but not in G1 and S phase in ALT-positive U2OS cells, indicating enhanced accumulation of SLX4 at telomeres. In contrast, co-treatment with HU and ATRi had no effect on the number of SLX4/TRF1 PLA foci in ALT-negative HeLa cells (Supplementary Figure 3A-3B). We appreciate the insightful feedback from the reviewer, which will significantly enhance the depth and quality of our study.

- On panel 1F adding CDC7 inhibitor to block origin firing seems to have increased PCNA ubiquitylation. CDC7i should decrease the number of active replication forks and the amount of PCNA on chromatin, so this is a surprising observation and should be discussed.

Thank you for pointing out this issue. We repeated the experiment and confirmed that treatment with CDC7 inhibitor PHA-767491 had no significant impact on PCNA monoubiquitination (please see the revised Figure 1G). In addition, the inhibitor did not influence the amount of PCNA on chromatin (cells were treated with HU, ATRi and PHA-767491 for only 1 hour) (please see the revised Figure 1G), likely because the firing of new replication origins induced by ATRi, which is blocked by the CDK7 inhibitor PHA-767491, only contributes to a small proportion of the total replication origins. This observation may warrant further investigation.

- On page 5 authors state "Upon activation, ATR activates its downstream kinases, including Chk1 and Wee1, to inhibit cell-cycle progression, suppress late origin firing, and stabilize stalled forks (Saldivar et al., 2017)." To my knowledge, there is no data indicating that ATR activates WEE1 in mammalian cells, and the cited review does not claim that either.

We apologize for any confusion. In *Xenopus laevis*, CHK1 (Xchk1) has been reported to phosphorylate Wee1 (Xwee1) at Ser547, which corresponds to Ser642 on human WEE1 (Lee *et al*, 2001; O'Connell *et al*, 1997). Our recent research has shown that in human cells, WEE1 phosphorylation at Ser642 upon DNA damage also relies on CHK1 and ATR (Zhu *et al*, 2023). We included these references in the revised manuscript to provide clarity. Thank you for bringing this to our attention.

Lee J, Kumagai A, Dunphy WG (2001) Positive regulation of Wee1 by Chk1 and 14-3-3 proteins. *Molecular biology of the cell* 12: 551-563

Nam EA, Zhao R, Glick GG, Bansbach CE, Friedman DB, Cortez D (2011) Thr-1989 phosphorylation is a marker of active ataxia telangiectasia-mutated and Rad3-related (ATR) kinase. *J Biol Chem* 286: 28707-28714

O'Connell MJ, Raleigh JM, Verkade HM, Nurse P (1997) Chk1 is a wee1 kinase in the G2 DNA damage checkpoint inhibiting cdc2 by Y15 phosphorylation. *EMBO J* 16: 545-554

Zhu X, Su Q, Xie H, Song L, Yang F, Zhang D, Wang B, Lin S, Huang J, Wu M *et al* (2023) SIRT1 deacetylates WEE1 and sensitizes cancer cells to WEE1 inhibition. *Nat Chem Biol* 19: 585-595

Prof. Jun Huang
Zhejiang University
Life Sciences Institute
388 Yuhangtang Road
Hangzhou, Zhejiang 310058
China

9th Feb 2024

Re: EMBOJ-2023-115612R
ATR limits Rad18-mediated PCNA monoubiquitination to preserve replication fork stability

Dear Dr. Huang,

Thank you for submitting your revised manuscript to The EMBO Journal. Two of the original referees have now assessed it once again (see comments below), and both of them are overall satisfied with your responses and revisions. We shall therefore be happy to accept the study for publication, after a final round of minor revision to incorporate the remaining presentational comments of referee 2, and to address the following editorial issues:

- Please change the "Supplementary" figures into Expanded View Figures: They should be renamed "Figure EV1/2/3" at all instances in the text, in the legends, and in the figure file.
- Please reorder the manuscript sections in the following order: Title page - Abstract & Keywords - Introduction - Results - Discussion - Materials & Methods - Data Availability - Acknowledgments - Disclosure & Competing Interests Statement - References - Figure Legends - Tables with legends - Expanded View Figure Legends.
- As we are switching from a free-text author contribution statement towards a more formal statement based on Contributor Role Taxonomy (CRediT) terms, please remove the present Author Contribution section and instead specify each author's contribution(s) directly in the Author Information page of our submission system during upload of the final manuscript. See <https://casrai.org/credit/> for more information.
- Please reduce the number of keywords on the abstract page to five (ideally choosing broad general terms).
- Please double-check to make sure to all relevant funding information in the manuscript is also entered into our submission system. (Missing in the system currently: National Natural Science Foundation of China [31961160725, 31730021, and 31971220 to J.H., 32270769, 31970664, and 31822031; Zhejiang Science Foundation for Distinguished Young Scholars [LR18C070001 to T.L.]; Fok Ying Tung Education Foundation, and Fundamental Research Funds for the Zhejiang Provincial Universities [2021XZZX039]).
- During our routine pre-acceptance checks, our data editors have raised the following queries regarding figures, data, and legends:
 - *Please note that in figure 6g, supplementary figure 2d; there is a mismatch between the annotated p values in the figure legend and the annotated p values in the figure file that should be corrected.
 - *Please note that scale bar and its definition are missing for figures 3g; 4a, c, e; 5f; 6d, f, supplementary figures 1h; 2a; 3a.
- Finally, please provide suggestions for a short 'blurb' text prefacing and summing up the conceptual aspect of the study in two sentences (max. 250 characters), followed by 3-5 one-sentence 'bullet points' with brief factual statements of key results of the paper; they will form the basis of an editor-written 'Synopsis' accompanying the online version of the article. Please also upload a synopsis image, which can be used as a "visual title" for the synopsis section of your paper. The image (maybe a compacted/simplified version of Figure 7?) should be in PNG or JPG format, and please make sure that it remains in the modest dimensions of (exactly) 550 pixels wide and 300-600 pixels high.

I am therefore returning the manuscript to you for a final round of minor revision, to allow you to make these adjustments and upload all modified files. Once we will have received them, we should be ready to swiftly proceed with formal acceptance and production of the manuscript.

Yours sincerely,

Hartmut Vodermaier

*** PLEASE NOTE: All revised manuscripts are subject to initial checks for completeness and adherence to our formatting guidelines. Revisions may be returned to the authors and delayed in their editorial re-evaluation if they fail to comply with the following requirements (see also our Guide to Authors for further information):

- size of the scale bars that are mandatory for all micrograph panels
- the statistical test used to generate error bars and P-values
- the type of error bars (e.g., S.E.M., S.D.)
- the number (n) and nature (biological or technical replicate) of independent experiments underlying each data point
- Figures may not include error bars for experiments with $n < 3$; scatter plots showing individual data points should be used instead.

3) Revised manuscript text (including main text, and figure legends for main and EV figures) has to be submitted as an editable text file (e.g., .docx format). We encourage highlighting of changes (e.g., via text color) for the referees' reference.

5) Point-by-point response letters should include the original referee comments in full together with your detailed responses to them (and to specific editor requests if applicable), and also be uploaded as an editable (e.g., .docx) text file.

9) Digital image enhancement is acceptable practice, as long as it accurately represents the original data and conforms to community standards. If a figure has been subjected to significant electronic manipulation, this must be clearly noted in the figure legend and/or the 'Materials and Methods' section. The editors reserve the right to request original versions of figures and the original images that were used to assemble the figure. Finally, we generally encourage uploading of numerical as well as gel/blot image source data; for details see: embopress.org/page/journal/14602075/authorguide#sourcedata

At EMBO Press, we ask authors to provide source data for the main manuscript figures. Our source data coordinator will contact you to discuss which figure panels we would need source data for and will also provide you with helpful tips on how to upload and organize the files.

In the interest of ensuring the conceptual advance provided by the work, we recommend submitting a revision within 3 months (9th May 2024). Please discuss the revision progress ahead of this time with the editor if you require more time to complete the revisions. Use the link below to submit your revision:

Link Not Available

Referee #2:

The authors have made a reasonable attempt to address the points raised by this reviewer (and, in my opinion, also those raised by the other reviewers), significantly improving the quality of the paper. Therefore, I would like to support its publication in the EMBO Journal, provided that the minor amendments described below are carried out.

Minor points

1. Figure 2E: The labelling for ATRi and HU is missing.
2. Page 12, last paragraph: MBP-SLX4-UBZ4-WT would be GST-SLX4-UBZ4-WT.

Referee #3:

The authors have addressed all of my concerns, I believe the article can now be published.

Referee #2:

The authors have made a reasonable attempt to address the points raised by this reviewer (and, in my opinion, also those raised by the other reviewers), significantly improving the quality of the paper. Therefore, I would like to support its publication in the EMBO Journal, provided that the minor amendments described below are carried out.

Thanks!

Minor points

1. Figure 2E: The labelling for ATRi and HU is missing.

Apologies for the oversight. The labeling for ATRi and HU has been added to Figure 2E.

2. Page 12, last paragraph: MBP-SLX4-UBZ4-WT would be GST-SLX4-UBZ4-WT.

Apologies for the error. This has been rectified. Thank you for bringing it to our attention.

Referee #3:

The authors have addressed all of my concerns, I believe the article can now be published.

Thanks!

Prof. Jun Huang
Zhejiang University
Life Sciences Institute
388 Yuhangtang Road
Hangzhou, Zhejiang 310058
China

26th Feb 2024

Re: EMBOJ-2023-115612R1
ATR limits Rad18-mediated PCNA monoubiquitination to preserve replication fork stability

Dear Prof. Huang,

Thank you for submitting your final revised manuscript for our consideration. I am pleased to inform you that we have now accepted it for publication in The EMBO Journal.

Yours sincerely,

Hartmut Vodermaier
